# Monitoring the binding and insertion of a single transmembrane protein by an insertase

Pawel R. Laskowski [1], Kristyna Pluhackova [1], Maximilian Haase[2], Brian M. Lang[1], Gisela Nagler[2], Andreas Kuhn[2] & Daniel J. Müller [1]✉

Cells employ highly conserved families of insertases and translocases to insert and fold proteins into membranes. How insertases insert and fold membrane proteins is not fully known. To investigate how the bacterial insertase YidC facilitates this process, we here combine single-molecule force spectroscopy and fluorescence spectroscopy approaches, and molecular dynamics simulations. We observe that within 2 ms, the cytoplasmic α-helical hairpin of YidC binds the polypeptide of the membrane protein Pf3 at high conformational variability and kinetic stability. Within 52 ms, YidC strengthens its binding to the substrate and uses the cytoplasmic α-helical hairpin domain and hydrophilic groove to transfer Pf3 to the membrane-inserted, folded state. In this inserted state, Pf3 exposes low conformational variability such as typical for transmembrane α-helical proteins. The presence of YidC homologues in all domains of life gives our mechanistic insight into insertase-mediated membrane protein binding and insertion general relevance for membrane protein biogenesis.

[1] Department of Biosystems Science and Engineering, ETH Zurich, 4058 Basel, Switzerland. [2] Molecular Microbiology, Biology Institute, Universität Hohenheim, 70599 Stuttgart, Germany. ✉email: daniel.mueller@bsse.ethz.ch

Transmembrane proteins play crucial roles in a substantial number of cellular processes. However, to function properly and avoid toxic aggregation their nascent polypeptide chains must be correctly inserted and folded into cellular membranes. While hydrophobic transmembrane segments of nascent polypeptides can spontaneously insert into cell membranes, the passage of hydrophilic polypeptide residues through the hydrophobic core of the membrane is thermodynamically unfavorable[1]. To overcome this free-energy barrier of translocation, the majority of membrane proteins require the assistance of insertases and/or translocases to catalyze their insertion and supervise their folding process[2–5]. There are two main insertion systems in bacteria, the Sec translocase (with SecYEG proteins) and the YidC insertase. The latter shares homology with Alb3 in chloroplasts, Oxa1 in mitochondria, and Get1 and Emc3 in the endoplasmic reticulum[6]. Being roughly five times more abundant in *Escherichia coli* compared to SecYEG[7], YidC can either transiently complex with SecYEG to form a holotranslocon or work independently[8]. YidC alone can insert transmembrane proteins including the subunit c of the $F_oF_1$-ATP synthase[9], the mechanosensitive channel MscL[10], the lactose permease LacY[5,11], and small and topologically simple proteins[12,13] such as the phage coat protein Pf3[14].

Unlike the Sec translocase, YidC does not form a transmembrane channel and its six transmembrane α-helices (TMH) form a hydrophilic groove, which is opened towards the cytoplasm to face polypeptides for insertion[15]. Whereas hydrophobic interactions assist transmembrane polypeptide segments to slide along TMH3 and TMH5 of YidC[16,17], the hydrophilic groove, which contains polar residues, including the highly conserved R366[18–21], assists polar periplasmic residues of the polypeptide to move deeply into and through the membrane[1]. The most flexible region of the insertase is formed by a cytoplasmic α-helical hairpin[19], which appears in all YidC homologues[22]. However, the role of the hairpin remains to be functionally understood in detail. Although the cytoplasmic α-helical hairpin is essential for inserting membrane proteins including Pf3 and M13 in *E. coli*, one cytoplasmic α-helix alone is sufficient to restore partial activity of YidC[23]. Moreover, deleting the cytoplasmic α-helical hairpin of YidC in *E. coli* and *Bacillus subtilis* severely decreases bacterial viability[18]. It is also hypothesized that once a transmembrane polypeptide has been inserted along YidC into the membrane, the free-energy barrier for translocating the polar polypeptide tail lowers[1]. However, the detailed understanding of how YidC inserts transmembrane polypeptides is incomplete. Particularly, the initial steps at which YidC binds polypeptides remain to be characterized.

Here, we illuminate how YidC binds and inserts substrates at high temporal and structural detail. To address this problem, we apply different atomic force microscopy (AFM)-based single-molecule force spectroscopy (SMFS) assays, Förster resonance energy transfer (FRET) spectroscopy, fluorescence correlation spectroscopy (FCS), and molecular dynamics (MD) simulations. The experiments show that YidC binds the Pf3 polypeptide within 2 ms at relatively low forces and doubles its binding strength to Pf3 within 52 ms. Our MD simulations and experiments corroborate a two-step binding and insertion mode. YidC first employs the cytoplasmic α-helical hairpin to bind the Pf3 polypeptide. Afterwards, the hydrophilic groove of YidC transiently interacts with Pf3 to insert it into the membrane. Along this pathway, the cytoplasmic α-helical hairpin does not only hand-over the polypeptide to the hydrophilic groove but also assists the insertion of the substrate. This binding and insertion of Pf3 is characterized by different alternative conformations of YidC-Pf3 complexes thus suggesting the polypeptide to follow different membrane insertion pathways.

## Results

**YidC insertase binds Pf3 polypeptide spontaneously.** To characterize substrate binding and insertion, functionally active YidC from *E. coli* was purified and reconstituted into POPE:POPG (3:1, w:w ratio) membranes (Supplementary Fig. 1a, b). Force–distance curve-based AFM (FD-AFM) imaging in buffer solution at room temperature showed that YidC distributed at lower density in membranes[24] (Supplementary Fig. 1c). To detect the interactions of the coat protein Pf3 with YidC, the C-terminal end of the purified Pf3 (Supplementary Fig. 1d) was covalently attached to the tip of an AFM cantilever by a flexible ≈9 nm long polyethylene glycol (PEG$_{27}$)-linker (Supplementary Fig. 2a). This attachment allowed the N-terminal end, from which Pf3 inserts into the membrane[15], to move freely. Using AFM-based SMFS in the height clamp mode[25], the functionalized tip was positioned ≈5–10 nm above a YidC membrane to investigate the binding of the unfolded Pf3 substrate to the insertase in force–time (FT) curves (Fig. 1a). The curves showed distinct interaction events in ≈2–4% of all cases ($n = 132/4,512$; Fig. 1b). Control experiments positioning Pf3-functionalized cantilevers above phospholipid membranes in the absence of YidC detected interaction events in <0.5% of all cases ($n = 4/910$), thus suggesting that the majority of the interaction events detected with YidC were caused by the insertase. We then extracted the lifetime and force of the binding events detected between YidC and Pf3 (Fig. 1c) and fitted their binned values with the Bell model[26] to estimate the lifetime of an average bond formed between YidC and Pf3 in the absence of any externally applied force (i.e., at thermal equilibrium) to be $t_0 = 0.32 \pm 0.25$ s. The procedure also estimated the width of the free-energy valley stabilizing the bound state to be $x_\beta = 0.57 \pm 0.27$ nm (Fig. 1c). Both values characterizing the bound state of Pf3 are comparable to values measured for the unfolding and extraction of single transmembrane α-helices from membranes proteins[27,28], thus indicating that the average bound state describes the fully inserted transmembrane α-helix of Pf3.

**YidC strengthens binding to Pf3 over time.** Next, we wanted to assess whether the binding force between YidC and Pf3 depends on the contact time between the insertase and the polypeptide. We again tethered the Pf3 polypeptide to the AFM tip via the PEG$_{27}$-linker and imaged YidC with FD-AFM (Fig. 2a). Upon imaging the YidC membrane, for each topographic pixel the FD-AFM approached and retracted the functionalized AFM tip to and from the membrane, while recording an approach and retraction FD curve, respectively (Fig. 2b, Supplementary Fig. 2b, c). Adjustments of the delay time between the approach and the retraction movement allowed to control the contact time between Pf3 and YidC. In this mode the approach FD curve described the Pf3 brought into contact with the membrane, whereas the retraction FD curve detected whether Pf3 bound to YidC. The FD-AFM imaging, which served to record large arrays (up to 262'144 FD curves per AFM topography) of SMFS experiments, is in the following named FD-AFM-based SMFS. A distance filter corresponding to the length of the stretched PEG-linker tethering Pf3 to the AFM tip allowed to select retraction FD curves detecting specific, single adhesion events of Pf3 to YidC[24] (Methods). The specific adhesion events detected in individual FD curves co-localized with the YidC molecules imaged in the FD-AFM topographs (Fig. 2a, b). To test the binding specificity of Pf3 tethered to the AFM tip, we used the functionalized AFM tips to image empty phospholipid bilayers and purple membrane, which contains only bacteriorhodopsin and lipids, by FD-AFM-based SMFS (Supplementary Fig. 3). This first set of control experiments ($n = 5$) showed negligible numbers of interactions, thus confirming the specificity of the interactions detected between YidC and Pf3. Next, we wanted to characterize whether the Pf3 polypeptide can insert into supported lipid membranes embedding YidC. Therefore, we added fluorescently

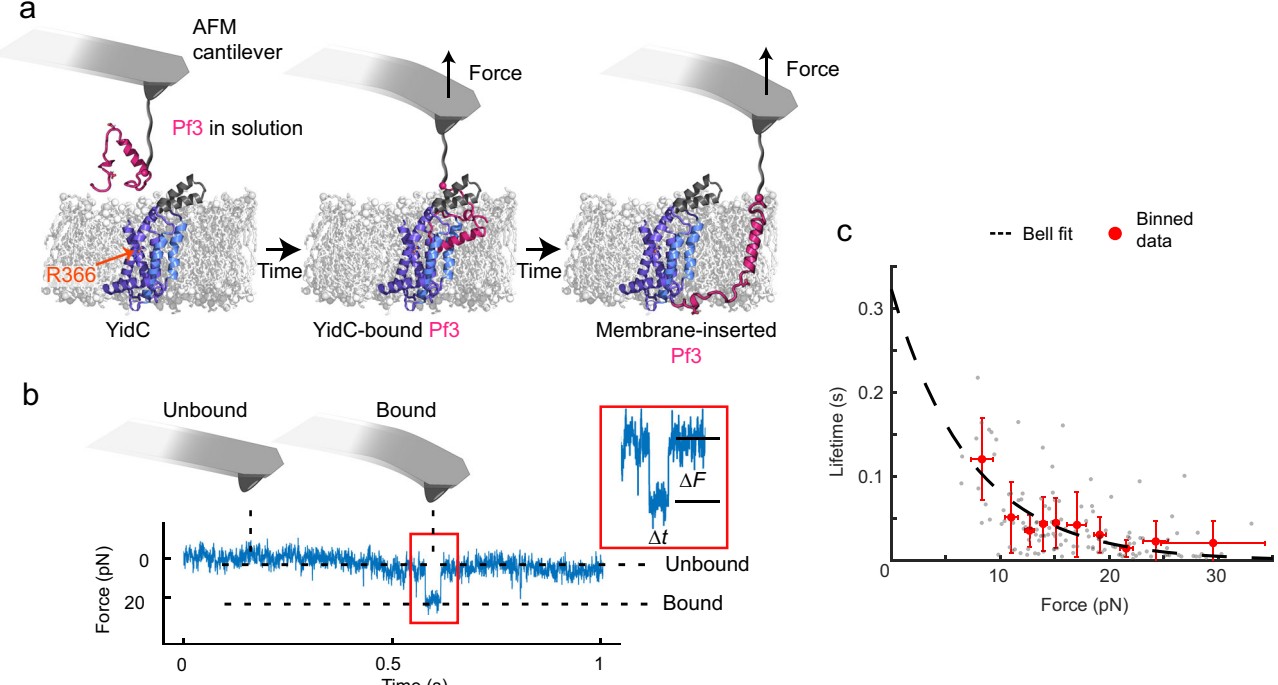

**Fig. 1 Spontaneous binding events of single Pf3 to wild-type (wt) YidC reveal life time and transition state. a** Schematic setup to detect interactions of YidC with Pf3 using AFM-based SMFS. Using AFM in the height clamp mode, the Pf3 polypeptide, which C-terminal end has been covalently tethered to the tip of the AFM cantilever (Supplementary Fig. 2), is kept in close proximity of ≈5–10 nm to a YidC containing membrane. If Pf3 (red) binds to YidC and/ or inserts into the membrane the $PEG_{27}$-linker tethering Pf3 to the tip stretches and the cantilever bends, thus detecting an interaction force. Highlighted structural regions of YidC are R366 (orange arrow), TMH3 and TMH5 (blue), and cytoplasmic α-helical hairpin (grey). **b** Example of a FT curve detecting a binding event of Pf3 to YidC. The force (ΔF) and time (Δt) of single binding events (inset) is extracted for analysis. **c** Analyzing the lifetime of single YidC-Pf3 binding events. Grey dots show individual data points ($n = 134$, where $n$ represents the number of binding events quantified) which were binned (red data points) and fitted with the Bell model[26] (black dashed line) to extract the lifetime of the bond in the absence of an external force (e.g., at thermal equilibrium) to be $t_0 = 0.32 \pm 0.25$ s (±95% confidence interval (CI)) and the transition state $x_\beta = 0.57 \pm 0.27$ nm (±95% CI) of the bond, which describes the distance Pf3 has to be pulled to separate from YidC. Error bars represent sd, which are centered at the mean value for each bin. Source data are provided as a Source Data file.

labeled Pf3 to the supported membranes and imaged the samples using a combined AFM and confocal microscopy setup (Supplementary Fig. 4). The merged AFM topographs and fluorescence images showed that Pf3 accumulated in membranes containing YidC. This second set of control experiments shows Pf3 not to insert into the supported phospholipid membranes in the absence of YidC.

After confirming that YidC in supported lipid membranes retains functionality and that our FD-AFM-based SMFS assay reproducibly detects specific (un-)binding events of YidC to Pf3, we varied the contact time between YidC and Pf3 in the experiments (Fig. 2c). In the absence of any further experimental adjustment, the minimum contact time between YidC and Pf3 was ≈2 ms, which we stepwise increased up to 502 ms (Methods). At 2 ms contact time, the average (un-)binding force between YidC and Pf3 equaled $29.5 \pm 12.4$ pN (mean ± sd). With increasing contact time, the (un-)binding forces increased and broadened their distribution suggesting stronger binding to be established. However, at 52 ms and 502 ms contact times the (un-)binding forces approached $47.6 \pm 17.3$ pN and $50.8 \pm 17.9$ pN, respectively, and their distributions were largely similar, thus indicating that they reached a stable, final state. To characterize whether the (un-)binding forces at both extended contact times represent the final membrane-inserted state of Pf3, we reconstituted Pf3 into POPE:POPG membranes (Methods) and measured the forces required to mechanically extract and unfold Pf3 from the membrane by FD-AFM-based SMFS (Supplementary Fig. 5). The distribution of the extraction and unfolding force and their mean value of $48.5 \pm 16.9$ pN showed no significant differences to

the distribution of the (un-)binding forces of $48.6 \pm 17.5$ pN collected at 52 ms and 502 ms contact times.

In summary, our single-molecule assay shows that the binding strength between YidC and Pf3 depends on the contact time. The distribution of the (un-)binding forces broadens with time and does not follow a normal distribution, which indicates that YidC and Pf3 establish multiple bonds / interactions along the substrate binding and insertion pathway. The relatively wide (un-)binding force distributions at 52 and 502 ms contact time do not differ, implying that the YidC-Pf3 complex reaches a final state latest at 52 ms. Moreover, the distribution of the forces required to mechanically (un-)bind Pf3 from YidC after contact times of ≥52 ms is similar to the distribution of the forces required to mechanically extract and unfold transmembrane Pf3 from the membrane. Both findings suggest that within a contact time of ≈52 ms YidC has completed the insertion of Pf3 into the supported lipid membrane, which is in agreement with previous FRET experiments that indicate YidC to insert Pf3 into proteoliposomes within 20 ms[29]. Our experiments thus describe that YidC binds Pf3 within ≈2 ms, thereafter strengthens the interactions with Pf3, and within 52 ms inserts Pf3 into the supported phospholipid membrane.

**YidC shows multiple substrate binding sites.** To understand which complexes may be formed during the YidC-mediated insertion of Pf3, we conducted multiscaling MD simulations[30] (Methods). First, the spontaneous binding of YidC to Pf3 was

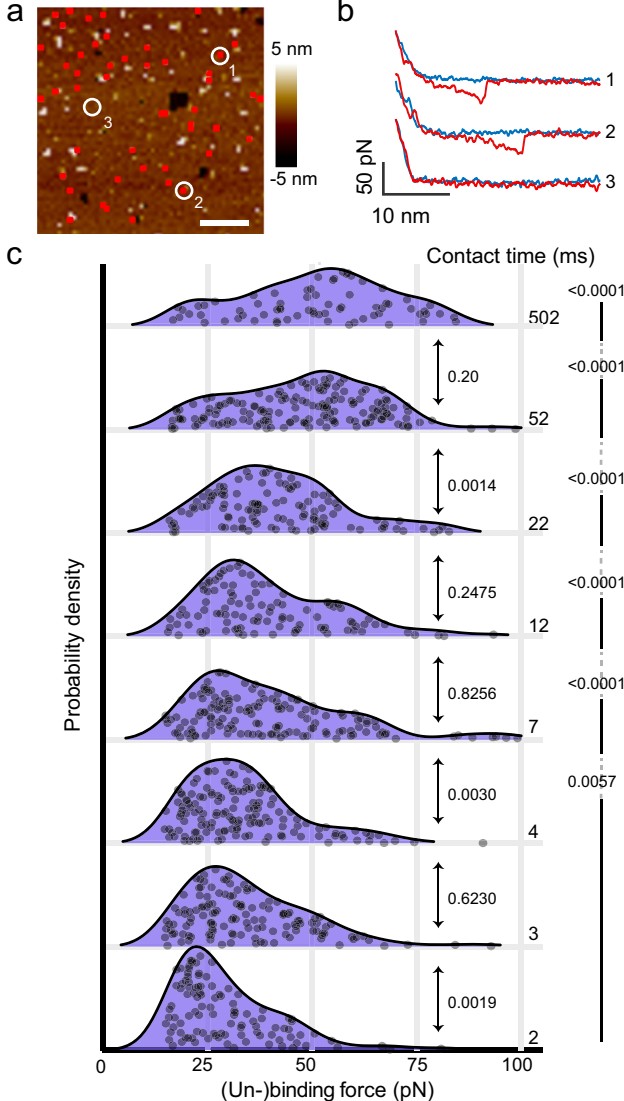

**Fig. 2 Binding of the YidC insertase to the Pf3 coat protein increases strength with time and saturates at 52 ms. a** FD-AFM topography of YidC reconstituted in phospholipid membranes. The topography has been recorded with an AFM tip that has been functionalized with Pf3 (Supplementary Fig. 2) to detect specific binding events to YidC. Red pixels show single binding events detected in the SMFS mode simultaneously conducted while recording the FD-AFM topography ($n > 5$, where $n$ represents the number of independent experiments). Scale bar, 200 nm. **b** Representative approach (blue) and retraction (red) FD curves as recorded for every pixel of the FD-AFM topography (**a**). Occasionally retraction FD curves detected single adhesion events at tip-membrane distances corresponding to the length of the PEG$_{27}$-linker that tethers the Pf3 polypeptide to the AFM tip. Shown are FD curves recording single (un-) binding events (top two) and no binding (bottom) of the Pf3 functionalized AFM tip with the YidC membrane. **c** Force profiles describing the (un-)binding of YidC and Pf3. With increasing contact time YidC strengthens binding to Pf3, which saturates at ≥52 ms. Numbers between force profiles depict $P$-values as calculated with a two sided Mann–Whitney U test between adjacent contact times (arrows) and relative to the 2 ms data set (right line). The (un-)binding force profiles were extracted from at least five independent experiments each detecting on average 15 single (un-)binding events of Pf3 and YidC. Grey dots show the raw data, individual (un-) binding forces ($n$ ranging from 69 to 158 data points) from which the probability density functions (lines and purple shaded areas) have been constructed. For a better display the data points of the (un-)binding forces have been randomly scattered along the $y$-axis. Source data are provided as a Source Data file.

studied at coarse-grained (CG) resolution (Supplementary Fig. 6). The simulations show Pf3 to bind the two cytoplasmic α-helices CH1 and CH2 of YidC in diverse orientations, each of which being stabilized by salt-bridges formed between the positively-charged lysines or arginines of YidC and the negatively-charged aspartic acids of Pf3 (Supplementary Fig. 6c, d). Out of these YidC-Pf3 complexes four were converted back to atomistic resolution[31] and re-equilibrated (Fig. 3a, Supplementary Fig. 7). Next, Pf3 was placed at different positions of the hydrophilic groove of YidC, which had previously been suggested to support an intermediate insertion step of the substrate[18], and equilibrated at coarse-grained resolution. Six different YidC-Pf3 complexes, which were stable over tens of μs, were then re-equilibrated atomistically for typically 1 μs each (Fig. 3b, Supplementary Fig. 8). The diversity of the stable complexes observed indicated several co-existing pathways along which Pf3 could approach the inserted state.

Using atomistic MD simulations, Pf3 was then repeatedly separated from each YidC-Pf3 complex and the (un-)binding forces recorded using FT curves (Fig. 3c, Supplementary Figs. 7 and 8). Separating Pf3, which had been partially inserted in the hydrophilic groove of YidC required higher (un-)binding forces than separating Pf3 from the cytoplasmic α-helical hairpin. In both cases the (un-)binding forces distributed broadly.

Independent of the YidC-Pf3 complexes formed and subsequently separated, the majority of the FT curves showed a second force peak following the maximum (un-)binding force. This second force peak described the mechanically dissociating N-terminus of Pf3 to rebind to the cytoplasmic α-helices of YidC (Supplementary Fig. 9, Supplementary Movie 1). As a reference, we simulated the (un-)binding of Pf3 adsorbed onto the phospholipid membrane or inserted into the membrane (Supplementary Fig. 10). While mechanically detaching Pf3 from the membrane surface required the lowest forces, (un-)binding Pf3 from the cytoplasmic α-helices of YidC required higher forces (Fig. 3d). Even higher (un-)binding forces were required to separate Pf3 from the hydrophilic groove or to extract transmembrane Pf3 from the membrane. Although we cannot distinguish, which of the observed YidC-Pf3 complexes used to simulate the interaction of YidC and Pf3 represent physiologically relevant states, the observations suggest that both the cytoplasmic α-helical hairpin and the hydrophilic groove of YidC play important roles in binding Pf3 and stabilizing the YidC-Pf3 complex. Furthermore, the spontaneous and weaker binding, which is first established by the cytoplasmic α-helical hairpin, and the stronger binding, which is formed by the hydrophilic groove, highlight a hierarchical mechanism of YidC towards binding and inserting Pf3.

The YidC-Pf3 complexes were then subjected to force distribution analysis (FDA)[32] to estimate which residues established either attractive or repulsive forces between YidC and Pf3 in both equilibration and steered MD simulations (Supplementary Fig. 11). In equilibration MD simulations of Pf3 bound to the cytoplasmic α-helical hairpin, Pf3 mainly attached to the positively charged arginines and lysines R384, R394, K401, and K416 of the cytoplasmic α-helices of YidC. These positively charged residues contributed to the highest (un-)binding force detected upon separating Pf3 from YidC (Supplementary Fig. 11a, c). When bound to the hydrophilic groove of YidC, Pf3 was stabilized by interactions formed with multiple residues in TMH2

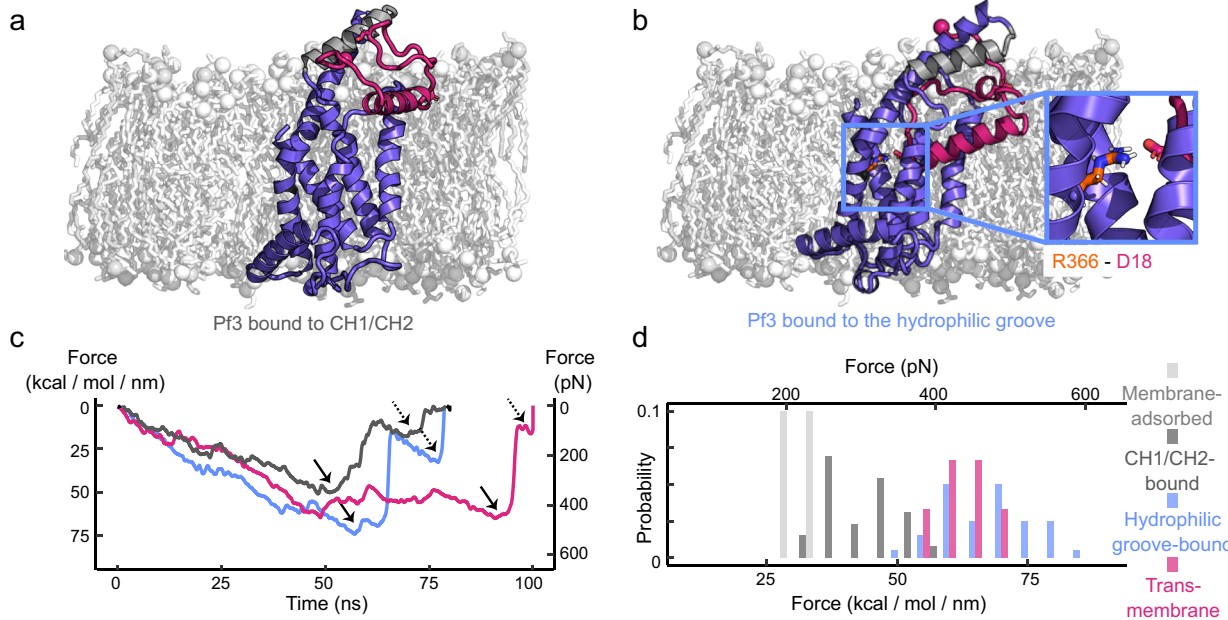

**Fig. 3 MD simulations reveal (un-)binding forces to depend on whether YidC binds the Pf3 polypeptide with the cytoplasmic α-helical hairpin or hydrophilic groove. a** MD simulation showing the cytoplasmic α-helices CH1 and CH2 (grey) of YidC to bind Pf3 (red). **b** MD simulation showing the hydrophilic groove (blue box) of YidC to bind Pf3 (red). The inset shows a salt bridge formed between R366 of YidC and D18 of Pf3. The highlighted interaction was observed in three out of five MD simulations. **c** Exemplary FT curves recorded upon mechanically separating Pf3 bound to YidC as revealed from steered MD simulations. FT curves describe the (un-)binding of Pf3 from CH1 and CH2 (dark grey), from the hydrophilic groove (blue), and the extraction of Pf3 from a membrane (red). Solid arrows indicate maximum (un-)binding forces and dashed arrows subsequently occurring weaker (un-)binding events. All FT curves are shown in Supplementary Figs. 7–10. **d** Distribution of maximum (un-)binding forces measuring the separation of Pf3 from CH1 and CH2 (dark grey, 287.2 ± 47.4 pN (mean ± sd), $n = 26$), from the hydrophilic groove (blue, 470.4 ± 57.9 pN, $n = 40$), and the extraction of Pf3 from the membrane (red, 439.8 ± 39.0 pN, $n = 6$). Reference maximum forces measure the separation of Pf3 adsorbed to phospholipid membranes (light grey, 223.0 ± 16.0 pN, $n = 6$, where $n$ refers to the number of quantified binding events. Snapshots along a typical FT curve are shown in Supplementary Fig. 9 with the pulling process being visualized in Supplementary Movie 1. Source data are provided as a Source Data file.

(including R366), TMH3 and TMH5 of YidC, which is in agreement with previous results[16]. Interestingly, several residues of the cytoplasmic α-helical hairpin (K401, R394, R384, D399, and M406) stabilized Pf3 in the hydrophilic groove (Supplementary Fig. 11b). Additionally, upon separating Pf3 from the hydrophilic groove of YidC, residues from the α-helix that connects TMH2 and CH1 of YidC exerted attractive forces to Pf3 and counteracted the externally applied pulling force (Supplementary Fig. 11d).

**YidC domains modulating substrate binding and insertion.** Our MD simulations highlight both the cytoplasmic α-helical hairpin and the hydrophilic groove of YidC to interact with the Pf3 polypeptide. To experimentally explore the functional role of both structural regions we decided to further characterize two YidC mutants. In the first mutant, ΔCH2 YidC, we deleted CH2 because this cytoplasmic α-helix was shown to impact the YidC function more than CH1[23]. In the second mutant, R366E YidC, the conserved positively-charged arginine R366 localizing deeply in the hydrophilic groove was replaced by a negatively-charged glutamic acid. We then determined the effect of each mutation on the insertion efficiency of Pf3 into YidC proteoliposomes using FCS (Fig. 4a, b, Supplementary Fig. 12). The fluorescent dye (Atto520) attached to the N-terminal end of Pf3 was quenched outside of the YidC proteoliposomes. However, if YidC translocated the N-terminal end of Pf3 to the inside of the proteoliposomes the fluorescence bursted, thus allowing to detect insertion events. While the insertion of Pf3 by wt YidC reached a plateau after 250 s, it was considerably reduced for both ΔCH2 YidC and

R366E YidC, indicating the impaired Pf3 insertion of both YidC mutants. Next, we applied FRET spectroscopy to characterize the average binding distance between YidC and Pf3 (Supplementary Fig. 13). While the acceptor (Atto647N) was attached to a cysteine in the cytoplasmic α-helical hairpin of wt YidC (wt YidC S405C) or of mutant R366E YidC (R366E YidC S405C), the donor (Atto520) was attached to the C-terminal region (47C) of Pf3. The experiments showed Pf3 and wt YidC to be at a distance of 4.6 nm, which extended to 5.8 nm for mutant R366E YidC, thus suggesting a different binding site. Compared to wt YidC, mutant R366E YidC showed only ≈25% of the FRET events within the same time period of 360 s. Taken together the results suggest that the YidC mutants use different and weaker binding sites to interact with Pf3 and show considerably reduced capacity to insert Pf3 into the membrane.

After having functionally characterized the YidC mutants, we used FD-AFM-based SMFS to characterize the (un-)binding of Pf3 from each YidC mutant at contact times ranging from 2 to 52 ms (Fig. 4c–e). For both YidC mutants we detected the binding of the substrate. Although individual FD curves showed single (un-)binding events similar to those detected upon (un-)binding Pf3 from wt YidC, the analysis of dozens of (un-)binding forces for each condition revealed significant differences. Intriguingly, at 2 ms ΔCH2 YidC (35.4 ± 16.8 pN; mean ± sd) established stronger interactions with Pf3 than wt YidC (29.5 ± 12.4 pN; Fig. 4c). By extending the contact time to 3, 4, and 12 ms, the mean (un-)binding forces of ΔCH2 YidC and wt YidC to Pf3 were similar. At 22 and 52 ms, however, the (un-)binding forces of Pf3 from ΔCH2 YidC were lower (41.0 ± 15.3 pN at 52 ms) than from wt YidC (47.6 ± 17.3 pN at 52 ms). The result thus shows that ΔCH2

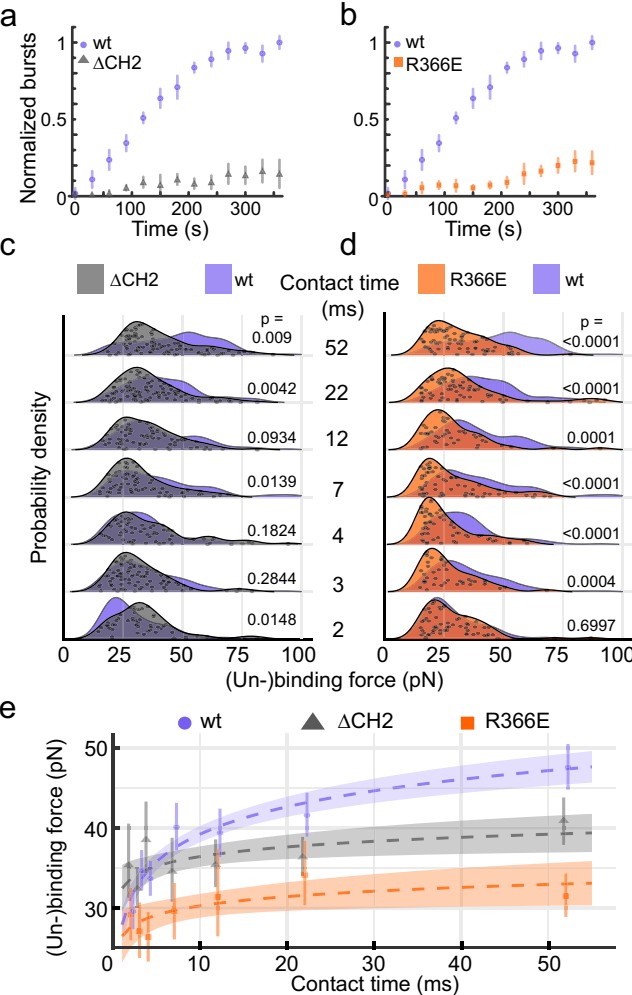

**Fig. 4 YidC mutated in the cytoplasmic region or hydrophilic groove binds Pf3 with different forces. a, b** Pf3 insertion into mutant ΔCH2 YidC (grey) or mutant R366E YidC (orange) and wt (purple) YidC proteoliposomes as measured by FCS. The Atto520 dye attached to the N-terminal end of Pf3 is quenched outside proteoliposomes and bursts fluorescence upon translocation via YidC into proteoliposomes. Data points represent means from 35 measurements and error bars sd. **c, d** (Un-) binding forces of ΔCH2 YidC (grey) or R366E YidC (orange) and the Pf3 polypeptide as detected by FD-AFM at different contact times (grey). Overlaid are (un-)binding forces of wt YidC and Pf3 (purple). Grey dots show data points from which the probability density functions (lines and shaded areas) were constructed. For better display data points were randomly scattered along the y-axis. Probability density functions from wt YidC were taken from Fig. 2c. Force–distance curves showing single unbinding events from either wt YidC, mutant ΔCH2 YidC or mutant R366E YidC are exemplified in Supplementary Fig. 16. **e** Multivariate linear regression (dashed lines) of wt (purple), R366E (orange) and ΔCH2 (grey) YidC built on the mean (un-)binding forces (dashed lines) of Pf3. The y-intercept equals 27.9 ± 2.2 pN (±95% CI), 32.5 ± 2.9 pN, and 26.5 ± 3.3 pN for wt, ΔCH2, and R366E YidC, respectively. Slopes of the (un-)binding forces equal 4.94 ± 0.91 pN s$^{-1}$, 1.74 ± 1.48 pN s$^{-1}$ and 1.66 ± 1.61 pN s$^{-1}$ for wt, ΔCH2 and R366E YidC, respectively. Circular, triangle and square data points give means (wt YidC, $n_{events}$ = 899; mutant R366E YidC, $n_{events}$ = 408; mutant ΔCH2 YidC, $n_{events}$ = 556) for each contact time. Shaded areas indicate 95% CI and error bars represent 95% CI of the means. Force distributions were statistically compared with a two-sided Mann–Whitney U-test showing P-values for each compared condition. Mean (un-)binding forces are summarized in Supplementary Table 1. Source data are provided as a Source Data file.

YidC can initiate the binding of Pf3 but over the time range tested cannot strengthen this binding to levels observed for wt YidC. In contrast, at 2 ms R366E YidC established (un-)binding forces to Pf3 of 29.1 ± 13.1 pN, which resembled the (un-)binding forces from wt YidC (29.5 ± 12.4 pN; Fig. 4d). Yet, R366E YidC failed to strengthen the (un-)binding forces for contact times ≥ 3 ms, which remained below those established by wt YidC. Finally, at extended contact times of 52 ms, the (un-)binding forces of Pf3 from R366E YidC (31.5 ± 12.3 pN) remained considerably below the (un-)binding forces detected for wt YidC. For example, at 52 ms contact time 48.7% of the forces describing the (un-) binding of Pf3 from wt YidC were >50 pN, whereas for ΔCH2 YidC only 22.2% of the forces were >50 pN and for R366E YidC only 10.5% were >50 pN. These differences in distributions of the (un-)binding forces of Pf3 from ΔCH2 YidC and R366E YidC suggest that they originate from different states. Particularly, since the initially (≈2 ms) established interactions of R366E YidC hardly strengthen with time, one may conclude that R366E YidC initially binds the Pf3 similarly to wt YidC, but fails to establish the interactions needed to insert the substrate into the membrane.

Next, we approximated the mean (un-)binding forces using multivariate linear regression (Fig. 4e). The regression supported that the forces depend on both the YidC type and the contact time. Moreover, the time-dependent strengthening of the (un-) binding force as estimated by the regression slope was much steeper for wt YidC compared to both YidC mutants. In conclusion, the FD-AFM-based SMFS results show that both the CH2 and the hydrophilic groove of YidC are required to strengthen the binding to the Pf3 polypeptide, while the FCS results show that YidC mutants either missing CH2 or having the hydrophilic groove mutated (R366E) can hardly insert Pf3 into the membrane. Together, both results suggest that YidC needs CH2 and hydrophilic groove to strengthen the initial binding to Pf3 and to insert the polypeptide into the membrane.

**Kinetics and thermodynamics of substrate-binding by YidC.** Next, we accessed the free-energy landscape parameters describing the binding and insertion of Pf3 by YidC. The parameters can be approximated by probing the (un-)binding forces of YidC-Pf3 complexes over a broader range of loading rates[33,34]. Hence, after allowing 2 ms for complex formation, Pf3 was separated from YidC at velocities ranging from 1–25 μm s$^{-1}$ using dynamic single-molecule force spectroscopy (DFS) (Fig. 5a). The most probable (un-)binding forces for each pulling velocity were plotted against the most probable loading rate (Supplementary Fig. 14) and fitted by the Bell–Evans model[33] to extract the transition rate $k_0$ of the YidC-Pf3 bond, which is reciprocal to the bond's lifetime, as well as the width of the free-energy valley $x_\beta$ and the height of the free-energy barrier $\Delta G$ stabilizing the bond (Table 1). We then measured the (un-) binding forces of the YidC-Pf3 complex at 52 ms contact time by DFS (Fig. 5b). The free-energy valley width of 0.81 ± 0.49 nm (±95% CI) detected at 52 ms was much narrower compared to the valley width of 2.0 ± 1.4 nm detected at 2 ms. This finding indicates that within the first ≈2 ms YidC initiates the binding of the Pf3 poly-peptide at higher conformational variability compared to the rather constrained membrane-inserted state reached after 52 ms. Moreover, the transition rate $k_0$ of the YidC-Pf3 interaction increased from 0.03 ± 0.28 s$^{-1}$ at 2 ms to 2.0 ± 9.1 s$^{-1}$ at 52 ms, which corresponds to life times of 33 s and 0.5 s, respectively. Consequently, the free-energy barrier stabilizing the complex decreased from 15.0 ± 9.3 $k_B T$ at 2 ms to 10.8 ± 4.5 $k_B T$ at 52 ms contact time. Finally, we compared how the ΔCH2 (Fig. 5c) and R366E (Fig. 5d) mutations affected the thermodynamic and kinetic parameters of the initial YidC-Pf3 binding (≈2 ms contact time). The widths of the free-energy valleys $x_\beta$ were very similar in wt (2.0 ± 1.4 nm), ΔCH2 (2.2 ± 2.0 nm), and

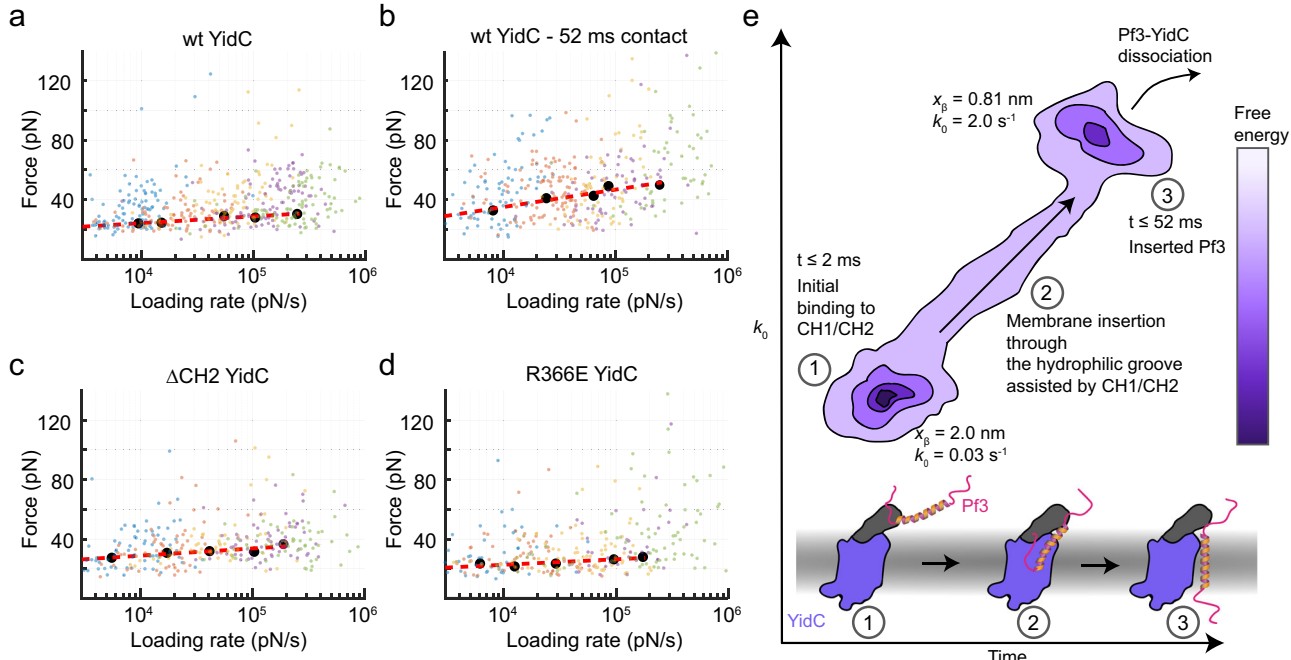

**Fig. 5 Free-energy landscape of YidC binding Pf3. a** (Un-)binding forces of wt YidC and Pf3 plotted against the loading rate. Data points represent single (un-)binding forces collected with SMFS at 2 ms contact time. **b** (Un-)binding forces of wt YidC and Pf3 collected at 52 ms contact time. **c** (Un-)binding forces of ΔCH2 YidC and Pf3 at 2 ms contact times. **d** (Un-)binding forces of R366E YidC and Pf3 at 2 ms contact time. Small dots represent single (un-)binding forces detected at pulling velocities of $1\,\mu m\,s^{-1}$ (blue), $3.1\,\mu m\,s^{-1}$ (orange), $6.3\,\mu m\,s^{-1}$ (yellow), $12.5\,\mu m\,s^{-1}$ (purple), and $25\,\mu m\,s^{-1}$ (green). Black larger dots represent most probable (un-)binding forces (Supplementary Fig. 14) and most probable loading rates calculated for each velocity using kernel density estimation. Bins were iteratively fitted using the Bell-Evans model[33] (dashed line) to estimate free-energy landscape parameters (Table 1). Each experiment was repeated at least five independent times. Total numbers of (un-)binding events in each plot for wt YidC were $n_{events} = 460$ (2 ms) and $n_{events} = 372$ (52 ms), for R366E YidC $n_{events} = 311$, and for ΔCH2 YidC $n_{events} = 322$. **e** Schematic free-energy landscape of YidC-mediated Pf3 binding and membrane insertion. The structural model at the bottom summarizes the mechanistic insight revealed in this study. YidC with its cytoplasmic α-helices are colored purple and grey, respectively. Pf3 is colored red and orange. Within 2 ms Pf3 binds to the cytoplasmic YidC surface in diverse conformations (1). Then within 52 ms, Pf3 migrates along multiple pathways (2), which involve the hydrophilic groove of YidC, to reach the membrane-inserted state (3). After these binding and insertion steps, the Pf3 polypeptide can dissociate from YidC. Source data are provided as a Source Data file.

**Table 1 Thermodynamic and kinetic parameters describing the binding established between wt, ΔCH2 or R366E YidC and Pf3.**

|  | wt YidC 2 ms contact time | wt YidC 52 ms contact time | ΔCH2 YidC 2 ms contact time | R366E YidC 2 ms contact time |
|---|---|---|---|---|
| $x_\beta$ (nm) | 2.0 ± 1.4 | 0.81 ± 0.49 | 2.2 ± 2.0 | 2.3 ± 1.7 |
| $k_0$ (s⁻¹) | 0.03 ± 0.28 | 2.0 ± 9.1 | 0.07 ± 0.84 | 0.03 ± 0.31 |
| $\Delta G$ ($k_B T$) | 15.0 ± 9.3 | 10.8 ± 4.5 | 14.2 ± 12.0 | 15.0 ± 10.3 |

$x_\beta$ and $k_0$ were extracted by fitting the DFS plots with the Bell-Evans model (Fig. 5a–d). $\Delta G$ was calculated using the Arrhenius equation (Methods). Errors in $x_\beta$ and $k_0$ describe 95% CI, while errors in $\Delta G$ were calculated by propagating errors in $k_0$.

R366E (2.3 ± 1.7 nm) YidC. The transition rates of Pf3 binding to YidC within 2 ms also showed the same order of magnitude for all three YidC variants ($k_0 = 0.03 ± 0.28\,s^{-1}$ for wt, $0.07 ± 0.84\,s^{-1}$ for ΔCH2, and $0.03 ± 0.31\,s^{-1}$ for R366E YidC).

Taken together, the thermodynamic and kinetic parameters that describe the initial binding (≈2 ms) of Pf3 are within the same range for wt, ΔCH2 and R366E YidC and indicate the corresponding YidC-Pf3 complexes to expose large conformational variabilities. The similarity of the parameters implies that the initially formed YidC-Pf3 interactions show similarities regardless of the mutation studied. However, the YidC mutants can hardly strengthen their interactions to the Pf3 polypeptide,

such as needed to efficiently insert the polypeptide into the membrane. The thermodynamic and kinetic parameters characterizing wt YidC-Pf3 (un-)binding at 52 ms contact time match well the parameters extracted from height clamp experiments ($k_0 = 3.1 ± 2.5\,s^{-1}$, $x_\beta = 0.57 ± 0.27\,nm$). It is also interesting to note that the width of the free-energy valley and the transition rate of Pf3 in the inserted state (≈52 ms) compare well to values reported earlier ($x_\beta ≈ 0.15–0.5\,nm$, $k_0 ≈ 0.3–4.9\,s^{-1}$) for single transmembrane α-helices of proteins spanning the membrane multiple times[28,35].

## Discussion

Here we have studied how the YidC insertase facilitates the insertion of the Pf3 polypeptide into the membrane. The interaction forces at which YidC initially binds Pf3 within ≈2 ms distribute widely and strengthen until reaching saturation at 52 ms, at which time YidC completed the insertion of Pf3 into the membrane. For all contact times tested (2–502 ms), the (un-)binding forces required to separate the YidC-Pf3 complex showed relatively broad distributions, which implies that YidC establishes various interactions with the Pf3 polypeptide. This observation is supported by MD simulations, which spot the formation of a variety of YidC-Pf3 complexes, whose conformations differ in how the cytoplasmic α-helical hairpin or/and the hydrophilic groove of YidC interact with the Pf3 polypeptide. Although we do not analyze these conformational changes in detail as this would go beyond the scope of our already rather extensive work, our observation of the various conformations fully support the

recently reported conformational changes of YidC upon Pf3 nascent chain binding[36] and translocation[37]. Furthermore, our simulations observe the cytoplasmic α-helices of YidC to contribute to the initial binding of the Pf3 polypeptide and that the positively-charged residues K401, R384 and R394 guide the polypeptide along multiple pathways to the hydrophilic groove (especially to R366), which transfers the polypeptide to the transmembrane state. The insights obtained from our simulations were experimentally confirmed upon characterizing mutant ΔCH2 YidC and mutant R366E YidC. In a further attempt we tested experimentally the role of residues K401, located in CH2 and R384 located in CH1 by FCS and FRET (Supplementary Fig. 15). This characterization of mutant R384E YidC and K401E YidC shows that they both lower the Pf3 insertion efficiency of YidC and affect the YidC-Pf3 binding conformations. Particularly mutant K401E YidC largely impairs the binding and insertion of Pf3, which suggests residue K401 as an interesting target for future functional studies of YidC. The manifold contributions of the cytoplasmic α-helix CH2 to the binding and insertion of Pf3 explain why deletion of the α-helix decreases cellular viability[23]. Also, previous studies observed the conserved positively charged R366 of the hydrophilic groove to be important for YidC to properly function[19] and that mutation R366E severely reduces Pf3 insertion[23]. Our FD-AFM-based SMFS experiments show that replacing R366 by a negatively charged glutamic acid does not affect initial substrate binding within the ms time scale, but hinders YidC to strengthen interactions such as occurring when inserting the substrate via the hydrophilic groove. Complementary, our FRET experiments show that mutant R366E YidC binds the substrate differently than wt YidC on the time scale of tens of seconds, and the FCS experiments show impaired substrate-insertion of the YidC mutant.

Previous studies investigated how Pf3, arrested at different translational positions, inserts into membranes via YidC[16]. The earliest contact to YidC was observed as soon as the Pf3 polypeptide exposed 25 residues from the ribosome and interacted with TMH3 and TMH5. Our studies identified this contact as the second binding step, which followed the initial binding of the N-terminal region of the Pf3 polypeptide by the cytoplasmic region of YidC. In this initial binding, the negatively charged amino-terminal region of Pf3 interacts electrostatically with the positively charged residues K401, R384, and R394 of the cytoplasmic α-helical hairpin of YidC. Thereafter, in a second step, the hydrophobic segment of the Pf3 polypeptide inserts along the TMH3 and TMH5 of YidC into the membrane and the positively charged R366 of the hydrophilic groove stabilizes either D7 or D18 of Pf3 deep in the membrane core. Both, simulations and experiments show that to guide the insertion process properly, YidC employs both the cytoplasmic α-helical hairpin and the hydrophilic groove.

To gain insight into how YidC initiates binding and facilitates insertion of the transmembrane polypeptide, we thermodynamically and kinetically characterized the YidC-Pf3 complex formed after 2 ms and 52 ms. We found that the free-energy valley stabilizing the initially bound state at 2 ms is wider compared to the narrow valley stabilizing the inserted state at 52 ms. Thus, to initially bind its substrate, YidC can form different complexes / conformations with Pf3, which are characterized by a high kinetic stability[38]. This observation is supported by our MD simulations, which show YidC to adopt various different conformations upon binding Pf3. After 52 ms at which Pf3, through assistance of YidC, has completed insertion into the membrane[29], the narrow free-energy valley describes the Pf3 to conformationally rigidify. In this inserted state, Pf3 shows thermodynamic and kinetic properties similar to single transmembrane α-helix from multispanning transmembrane α-helical proteins[28,35]. Additionally, the (un-)binding forces required to mechanically separate Pf3 after 52 ms from the YidC membrane

are similar to the forces required to mechanically extract reconstituted Pf3. Interestingly, with increasing contact time from 2 to 52 ms, the lifetime of the YidC-Pf3 complex decreases ≈30 – 60-fold to ≈0.5 s, which suggests that once Pf3 has been inserted in the membrane the complex becomes kinetically less stable. Such instability may be needed to support the dissociation of the membrane-inserted Pf3 from the YidC insertase. Even though experiments in free proteoliposomes suggest this dissociation to take place at time scale of ≤20 ms[29], this process may be slowed down in supported lipid bilayers[39].

Based on our experimental findings and simulations we contour a free-energy landscape, a commonly used approximation to describe the (un-)binding of ligand-receptor pairs or protein (un-)folding[40,41]. We describe the binding and insertion of polypeptides by YidC via (at least) two free-energy valleys (Fig. 5e). The first valley describes the initial binding of the Pf3 polypeptide by YidC, which occurs within very short time ranges (≈2 ms) and is relatively wide, which suggests that the YidC-Pf3 complexes can adopt various conformations. This initial binding, which is mainly facilitated by the cytoplasmic α-helices of YidC, has a long lifetime of ≈33 s such as needed to prevent the YidC-Pf3 complex to dissociate. The fact that YidC-Pf3 complexes that describe the initial binding of the substrate can show many different conformations of long lifetime suggests the cytoplasmic α-helices of YidC to work like an efficient 'flytrap' to catch (bind) the substrate from the cytosol. In a next step, the Pf3 polypeptide with the help of the cytoplasmic α-helices is guided to the hydrophilic groove. After binding the hydrophilic groove, the Pf3 reaches the free-energy valley of the membrane-inserted state within 52 ms. This insertion is supported by electrostatic attractions between negatively-charged residues of the Pf3 N-terminus and R366 of the hydrophilic groove of YidC. However, the cytoplasmic α-helical hairpin also contributes to this process. Once the Pf3 polypeptide is inserted and folded into the membrane, the N-terminal region will move from the groove-bound state to the periplasm[16]. Compared to the free-energy valley describing the initial substrate-binding, this second valley of the membrane inserted state is much narrower thus providing Pf3 less conformational variability, such as described for transmembrane α-helices[28,35]. In addition, the free-energy barrier stabilizing the complex is lower, suggesting that Pf3 in the inserted state may still be associated with YidC and the lower free-energy barrier supports the dissociation of the complex. Together, the results suggest a hierarchical mode of interaction and insertion of Pf3 by YidC. Given the shared homology of the members of the insertase family of different organisms, it will be interesting to learn which commonalities the other members show in inserting transmembrane proteins and in which details they differ. Ultimately this will contribute to a mechanistic understanding of how insertases work.

## Methods

**YidC purification and reconstitution**. All YidC had a C-terminal His-tag with the wild-type cysteine at position 423 mutated into a serine. By site-directed mutagenesis two mutants were generated. In ΔCH2 YidC mutant residues 399–415 are deleted. Vector pMS119EH in *E. coli* C43 strain was used as expression system. At an optical density of 0.5 the cells were induced with 1 mM isopropyl-β-D-thiogalactopyranosid (IPTG) and incubated for another 2 h. The cells were harvested and disrupted in One Shot Cell Disruptor (Pressure Biosciences Inc.) at a pressure of 1.3 kBar. The cell debris was separated at 6000 g at 4 °C and the membrane fractions were sedimented at 50,000 g for 1.5 h. Membranes were solubilized overnight at 4 °C in solubilization buffer (300 mM NaCl, 15 mM imidazole, 20 mM Tris-HCl, 1% n-Dodecyl β-D-maltoside (DDM), pH 7.5). Non-solubilized components were separated by centrifugation at 50,000 g at 4 °C for 1 h. The supernatant was incubated with Ni$^{2+}$-NTA at 4 °C for 1 h and washed with wash-buffer (300 mM NaCl, 30 mM imidazole, 0.05% DDM, 20 mM Tris-HCl pH 7.5). 1 ml of each fraction was eluted with elution-buffer (300 mM NaCl, 300 mM imidazole, 0.05% DDM, 20 mM Tris-HCl, pH 7.5). Next, YidC was reconstituted into lipid vesicles. Lipids 1-palmitoyl-2-oleoyl-sn-glycerol-3-phosphoethanolamine (POPE) and 1-palmitoyl-2-oleoyl-sn-glycerol-3-phospho-(1′-rac-glycerol) (POPG) were used in powder form, mixed in 3:1 ratio (w:w) and dissolved in dichloromethane. The liquid was removed by a rotary evaporator at 50 mbar which formed a lipid film. The film was dried under vacuum for 6 h and resuspended in water

with a concentration of 10 mg ml⁻¹. The resuspended lipid was diluted with a buffered solution (300 mM NaCl, 40 mM Tris-HCl, pH 7.5) to a concentration of 5 mg ml⁻¹. The purified YidC was mixed with lipids in a 5:1 (lipid:protein, w:w) ratio. The YidC-lipid solution was extruded 10 times through a 0.4 µm nitrocellulose membrane. To remove the remaining detergent (DDM), the proteoliposomes were incubated overnight with biobeads (BIO-RAD). Ultracentrifugation at 40,000 g separated the non-reconstituted YidC and proteoliposomes.

**Purification of Pf3 coat protein.** Pf3 was modified by site-directed mutagenesis with a single cysteine mutation 47C for FRET and SMFS and at 3C for FCS insertion experiments. Pf3 was expressed from *E. coli* BL21 cells, which were lysed at 1 kBar in One Shot Cell Disruptor. The cell debris was centrifuged at 20,000 g for 1 h and the supernatant was diluted in buffered solution (5% isopropanol and 0.1% trifluoroethanol at (v:v = 1:4), 20 mM Tris-HCl, pH 8.0) to the crude extract. The crude extract was purified by reversed phase chromatography and then further purified by size exclusion chromatography using Superdex 200 and Superdex 75 columns.

**Reconstitution of Pf3.** Purified Pf3 was reconstituted into POPE:POPG (3:1 ratio) following the protocol above describing the reconstitution of YidC. Briefly, POPE and POPG in powder were mixed in 3:1 ratio (w:w) and dissolved in dichloromethane. The liquid was removed by a rotary evaporator at 50 mbar which formed a lipid film. The film was dried under vacuum for 6 h and resuspended in water with a concentration of 10 mg ml⁻¹. The resuspended lipid was diluted with a buffered solution (300 mM NaCl, 40 mM Tris-HCl, pH 7.5) to a concentration of 5 mg ml⁻¹. The purified Pf3 was mixed with lipids in a 5:1 (lipid:protein, w–w) ratio. The Pf3-lipid solution was extruded 10 times through a 0.4 µm nitrocellulose membrane. To remove the DDM, the proteoliposomes were incubated overnight with biobeads (BIO-RAD). Ultracentrifugation at 40,000 g separated non-reconstituted Pf3 from proteoliposomes.

**Atomic force microscopy (AFM) tip functionalization.** To separate specific from non-specific interactions, our AFM-based SMFS experiments were designed in a way that specific unbinding events detected in force curves can be clearly discerned in distance from possible non-specific interactions[42]. We have thus tethered a PEG₂₇-linker, which in the stretched conformation is ≈9 nm long, to the AFM tip. At the free end of the linker we have covalently attached the C-terminal end of the Pf3 polypeptide. The protocol for functionalizing the AFM tip with this PEG₂₇-linker-Pf3 system was as following: AFM probes used had a nominal spring constant of 0.07 N m⁻¹ and resonance frequency of 25 kHz in water (AC40, Bruker). Fresh cantilevers were cleaned for 10 min in ultraviolet radiation and ozone (UV-O cleaner, Jetlight) and submerged overnight in ethanolamine solution (3.3 mg of ethanolamine (Sigma-Aldrich) in 6.6 ml of dimethyl sulfoxide (DMSO, Sigma-Aldrich)). Next, the cantilevers were washed three times in DMSO, rinsed in a gentle stream of ethanol and dried with nitrogen. The cantilevers were then immersed for 2 h in solution of maleimide-PEG₂₇-N-hydroxysuccinimide (NHS) (1 mg of maleimide-PEG-NHS in 500 µl chloroform and 30 µl triethylamine). PEG-coated cantilevers were incubated three times in fresh chloroform for 10 min and dried with air. Next, the cantilevers were incubated for 4 h in a solution of 100 µl of 2 µM Cys-modified Pf3 in PBS buffer supplemented with 10% isopropanol, which was premixed with 2 µl of 100 mM EDTA (pH 7.5), 5 µl of 1 M HEPES (pH 7.5), 2 µl of 100 µM TCEP hydrochloride and 2 µl of 1 M HEPES (pH 9.6). The cantilevers were washed in PBS buffer with 10% isopropanol and stored in this solution at 4 °C up to one week.

**Sample preparation for AFM.** Sample supports were round mica disks (≈0.05 mm thickness, 9.5 mm diameter) punched using a 'punch and die' set (Precision Brand Products). Mica discs were glued on slightly larger Teflon discs. Teflon discs were glued onto metal discs which were later magnetically attached to the AFM stage. Reconstituted YidC was diluted 100 times in imaging buffer (150 mM KCl, 20 mM HEPES, pH 7.4) and adsorbed onto freshly cleaved mica for 30 min. The sample was then washed ten times using 50 µl of imaging buffer. Buffers were prepared using nanopure water (18 MOhm cm⁻¹) and analytical grade chemicals. All preparation steps were performed at room temperature.

**FD-AFM-based single-molecule force spectroscopy (SMFS).** The AFM (Nanoscope Multimode 8, Bruker) was equipped with a 120-µm piezoelectric scanner and operated in the FD-AFM mode (Force-Volume, Nanoscope software v9.1, Bruker). Membrane patches embedding YidC were imaged with Pf3-functionalized cantilevers at imaging force of 100 pN, 40 nm ramp size, 64 or 128 pixels per line and 512 points per FD curve. The vertical pulling velocity of the cantilever was set to 3–4.5 µm s⁻¹ for the experiments in Figs. 2 and 4. Upon approaching and retracting the cantilever to a membrane at this velocity and imaging force, the AFM tip contacted the membrane for ≈2 ms. To increase this contact time, the retraction of the cantilever was delayed by 1–500 ms after the approaching cantilever reached the imaging force. This delay time added to the minimal contact time of ≈2 ms thus increasing the total contact time between Pf3 and YidC membranes. For DFS (Fig. 5) the pulling velocities were set to 1 µm s⁻¹, 3.1 µm s⁻¹, 6.3 µm s⁻¹, 12.5 µm s⁻¹ or 25 µm s⁻¹. To provide a constant contact time

in dynamic force microscopy (DFS) experiments throughout all pulling velocities, delay times of 0.7 ms (6.3 µm s⁻¹), 1 ms (12.5 µm s⁻¹) or 1.2 ms (25 µm s⁻¹) were added. AFM cantilevers were calibrated using thermal tuning and by ramping on solid surfaces. All experiments were performed at room temperature and repeated at least five times.

To mechanically extract and unfold single Pf3 reconstituted into POPE:POPG lipid membranes, a non-functionalized AFM tip was approached to and pushed against the membrane surface until reaching a force of 150–200 pN. After 75–100 ms allowing the polypeptide to attach to the AFM tip, the cantilever was retracted at a velocity of 3.1 µm s⁻¹. During approach and withdrawal FD curves were recorded. In ≈0.34% of all cases (n = 127/37896), the retraction FD curve showed an adhesion force peak indicating the extraction of a Pf3 from the membrane.

To collect force-time (FT) curves in the height clamp mode, the AFM (Bioscope Resolve, Bruker) was operated in the FD-AFM mode (Peak Force QNM mode, NanoScope 9.4R3 software, Bruker). The AFM tip was approached to the sample with ≈100 pN imaging force and then retracted ≈5–10 nm away from the surface. Next, the cantilever was kept at a constant height for ≈1–2 s while recording the cantilever deflection. After that time, the cantilever was retracted by ≈50 nm and the measurement was repeated.

**Combined AFM and confocal microscopy.** An AFM (BioScope Resolve, Bruker) was installed on the stage of an inverted confocal microscope (LSM 800, Carl Zeiss) and operated in the FD-AFM mode. The AFM was equipped with a 100 × 100 × 15 µm (x, y, z) piezoelectric scanner. Purified Pf3 was fluorescently labeled with maleimide-Atto488 dye (Sigma-Aldrich) using the same chemical reaction used to functionalize the cantilever. An excess of unbound dye was removed with a 3 kDa centrifugation filter (Amicon) and the completeness of chemical reaction was confirmed with size exclusion chromatography (Akta). Reconstituted YidC membranes were adsorbed on mica discs glued on glass slides. After adsorption of the membranes, a fluorescence dye Atto647 was non-specifically adsorbed to mica to block the non-specific adsorption of the specific fluorescence dyes[43]. Then, the sample was incubated for 30 min with fluorescently labeled Pf3 to the fluorescent signal from Pf3 inserted into YidC membranes. The sample was rinsed 10 times with buffer solution to remove the unbound dye. Confocal images were collected using a 10 mW, 488 nm laser at 3–10% power and a 1 airy unit pinhole. Optical images were acquired with a 63× water immersion lens (421787-9970-799 objective, NA 1.20, Carl Zeiss).

**Analysis of force-distance (FD) and force-time (FT) curves.** FD curves were extracted and analyzed for specific adhesion events using an in-house developed MATLAB code. To exclude non-specific adhesions events, all FD curves from YidC-Pf3 interactions were filtered for adhesion events occurring at distances 5–25 nm from the contact point, which corresponds to the length of PEG₂₇ linker complexed with the Pf3 polypeptide. To select single YidC-Pf3 interactions, only FD curves showing a single specific adhesion event of a force being five times higher than the sd of the baseline noise (≈15 pN) were considered for analysis. FD curves extracting reconstituted Pf3 were filtered for adhesion events occurring at distances 10–25 nm from the contact point, which corresponds to the contour length of the fully unfolded and stretched Pf3 polypeptide (≈15 nm).

FT curves from height clamp experiments were smoothened with a running average and filtered for single binding events, which started and ended at the baseline. FT curves which showed fluctuating of more than 5 pN during a binding event were excluded from analysis. All graphs were plotted in R version[44] and MATLAB. (Un-)binding force profiles were statistically compared with Mann–Whitney U tests.

**Extraction of kinetic parameters.** Estimation of kinetic and thermodynamic parameters of the (un-)binding events was performed by fitting the Bell (Fig. 1) and Bell–Evans (Fig. 5) model with Curve Fitting Tool in MATLAB (Mathworks). Bins from force clamp experiments (Fig. 1) were fitted to the Bell model[26] using the equation:

$$t(F) = t_0 \, e^{\frac{x_\beta F}{k_B T}} \tag{1}$$

where $t(F)$ is bond lifetime as a function of force, $t_0$ is the bond lifetime in equilibrium, $x_\beta$ is the width of the energy valley, $k_B$ is the Boltzmann constant and $T$ the temperature.

DFS datasets were divided by five pulling velocities (Fig. 5). The most probable rupture force and the most probable loading rate were estimated with kernel density estimation and the values were fitted to the Bell–Evans model[33] using Levenberg–Marquardt algorithm with weights proportional to the number of observations in each bin:

$$F(LR) = \left(\frac{k_B T}{x_\beta}\right) \ln \frac{LR \, x_\beta}{k_0 \, k_B T} \tag{2}$$

where $F(LR)$ is (-un)binding force as a function of loading rate and $k_0$ is zero-force transition rate.

The free-energy $\Delta G$, which describes the free-energy difference between the unbound and the bound state, was calculated using the Arrhenius equation:

$$\Delta G = -k_B T \ln(\tau_D k_0) \tag{3}$$

where $\tau_D$ is the diffuse relaxation time, estimated to have a value of $10^{-5}$ s, as published[45,46]. Errors in $\Delta G$ were calculated by propagation of errors in $k_0$.

**Fluorescence correlation spectroscopy (FCS).** The measurements were performed on a self-built FCS instrument with the confocal Olympus IX71 microscope equipped with a water immersion objective (UPlanSApo 60x, N.A. 1.2, Olympus). The molecules in the measurement volume were excited with a 491 nm laser reduced from 50 mW to 115 µW by a cleanup filter. The detection of the single fluorescent photons was performed by an avalanche photodiode (SPCM-AQRH-14, Excelitas Technologies) after filtering the excitation wavelength from the emission wavelength by a dichroic beam splitter (zt488RDC; AHF Tübingen). The single photons were then processed by a time-correlated single photon counting (TCSPC) card (SPC153; Becker & Hickl). The signals were recorded for 360 s and only photon bursts with a minimum average intensity of 50 counts/bin were examined. The diffusion time of at least 40 ms was determined for the proteoliposomes with YidC and for Pf3 inserted in proteoliposomes. YidC proteoliposomes were diluted in 40 µl of buffered solution (300 mM NaCl, 20 mM Tris-HCl, pH 7.5). The sample was placed on the glass slide and mixed with 10 µl of Pf3 3C labeled with Atto520 (5% isopropanol, 300 mM NaCl, 20 mM Tris-HCl, pH 7.5). YidC and labeled Pf3 had a final concentration of 1 nM. The reaction mixture was supplemented with 100 mM KI used as fluorescence quencher such that only fluorescence from inserted Pf3 is protected. The measurements were recorded for 360 s. The extracted data was plotted in Origin.

**Förster resonance energy transfer (FRET) spectroscopy.** FRET spectroscopy was carried out on the FCS setup described above to characterize the binding of Pf3-47C to YidC S405C. In addition to the avalanche photodiode for the emission wavelength at 535 nm, another avalanche photodiode was used to record at the emitted acceptor wavelength at 635 nm in order to detect the acceptor signal separately from the donor signal. The signals were correlated by a TCSPC (DCP 230 card, Becker & Hickl) burst analyzer software (Becker & Hickl). 40 µl of proteoliposome solution (300 mM NaCl, 20 mM Tris-HCl, pH 7.5) with 1 nM YidC S405C labeled with Atto647N was placed on a cover slide and mixed with 10 µl of 1 nM Atto520 labeled Pf3-47C (5% isopropanol, 300 mM NaCl, 20 mM Tris-HCl, pH 7.5). Both proteins had a final concentration of 1 nM. Fluorescent signals of both the donor and the acceptor wavelength were recorded and analyzed for 360 s.

**Structure preparation for molecular dynamics (MD) simulations.** The periplasmic P1 domain of YidC, which is not essential for its function[47], was removed to reduce the simulation system size. The structurally unresolved, non-conserved TMH1 was omitted. Thus, the YidC structure started with the conserved amphipathic α-helix EH1, essential for the functionality of the insertase. The terminal Val328 and Tyr532 were uncharged because they do not represent native protein termini. All side chains were protonated according to their standard protonation states at neutral pH. Pf3 used in simulations had residues Met1 to Phe44 and charged termini. As Pf3 is partially α-helical in the preinserted state (≈40% of the peptide is α-helical, corresponding to 18 residues[48], the hydrophobic part of the peptide (I19-I36) was imposed to be α-helical in all coarse-grained (CG) simulations. The membrane composition equaled to that in the experiments, i.e., we have mixed POPE and POPG in 3:1 molar ratio and solvated the membrane in 150 mM NaCl solution. Full hydration was achieved by lipid:water ratios of about 1:90 in the equilibration simulations and 1:230 in pulling simulations.

**MD simulations.** All molecular dynamics simulations were performed at 25 °C using GROMACS 2018[49]. In CG resolution the polarizable variant of the Martini force field version 2 was used[50–52] to capture the electrostatic interactions[53]. Selected CG structures were then converted to atomistic resolution described by the CHARMM36m force field for proteins[54], CHARMM36 force field for lipids[55] and the TIP4p water model[56] using the routine *backward*[31], re-equilibrated and pulled apart. In all-atom simulations, the well-tested simulation parameters for the CHARMM36 force field were used[57,58] and in Martini simulations the recommendation of de Jong at all applied[59]. For more details, see Supplementary Methods.

**Analysis of MD simulations.** Most of the analysis was performed using standard GROMACS tools and in-house written R scripts[44]. The force distribution analysis (FDA)[32] was utilized to find YidC residues important for binding of Pf3. In FDA analysis residue-wise forces between YidC and Pf3 were estimated and saved as scalars for each trajectory frame. In detail, FDA was performed on the last 10 ns of equilibration simulations in order to reveal residues important for bound states, and was averaged over all structures belonging either to the hydrophilic groove-bound or to the CH1/CH2-bound complexes. Moreover, in order to pinpoint residues stabilizing Pf3 at YidC at the maximal external pulling force, FDA was used to estimate residue-wise forces between YidC and Pf3 at the time of the

maximum external pulling force and averaged over all pulling simulations and structures belonging either to the hydrophilic groove-bound or to the CH1/CH2-bound complexes. PyMOL[60] was used for visualization.

**Reporting summary.** Further information on research design is available in the Nature Research Reporting Summary linked to this article.

## Data availability
The data that support the findings of this study are available from the corresponding author upon reasonable request. Source data are provided with this paper.

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

## Acknowledgements

We are thankful to D. Alsteens, G. Fläschner, K. Kasuba, and F. Wilhelm for helpful discussions. We thank N. Blaimschein for assistance in Pf3 purification. The work was supported by the Swiss National Science Foundation (SNFS, grant 2053020_160199/1) and the NCCR Molecular Systems Engineering. K.P. acknowledges financial support of the German Research Foundation (DFG) (Forschungsstipendium PL 853/1-1). The authors gratefully acknowledge the Gauss Centre for Supercomputing e.V. (www.gauss-centre.eu) for supporting this project (project ID pr27wi) by providing computing time on the GCS Supercomputer SUPERMUC-NG at Leibniz Supercomputing Centre (www.lrz.de).

## Author contributions

D.J.M., P.R.L., and A.K. conceived the project. G.N. purified proteins. P.R.L., M.H., and G.N. performed experiments and P.R.L., M.H., and B.M.L. analyzed the results. K.P. designed, conducted, and analyzed MD simulations. All authors discussed and wrote the manuscript.

## Competing interests

The authors declare no competing interests.
