## [Peer Review File · Nature Communications]

Monitoring the binding and insertion of a single transmembrane protein by an insertaseReviewers' Comments:

Reviewer #1:

Remarks to the Author:

Using a variety of sophisticated methods, Lakowski et al. present data that reveal the pathway of pf3 insertion into lipid membranes guided by the YidC insertase. The paper is of fundamental importance and should be published--eventually. The fundamental problems are the organization and presentation of the data. Not until the reader reaches the end of the paper does s/he really come to understand the question being examined, the strategy being used to answer it, and the answer. The interested but non-expert reader is likely to abandon the paper wondering what the point of it is. There are five main figures and 17(!) supplementary figures that are exhausting to read even for experts. In short, the paper is a mess and a nightmare to read. Reading the paper is a slog. One thing that would help is a closing figure that summarizes the essential findings for the non-expert reader.

The paper should be reconsidered after the authors dramatically improve the presentation of the material with the help of a native english speaker.

Reviewer #2:

Remarks to the Author:

The manuscript by Laskowski et al. provides mechanistic details on the binding and insertion of a single-pass transmembrane protein (Pf3) mediated by YidC, a bacterial insertase. The authors use a combination of various biophysical, computational, and biochemical techniques to decipher the Pf3 binding and insertion process by YidC. Using a combination of AFM and SMFS, the lifetime and strength associated with binding events were characterized. The single-molecule assays show that the binding strength between YidC and Pf3 depends on the contact time. They conclude that Pf3 binds YidC within less than 2 ms and inserts into the membrane within 52 ms. Their coarse-grained (Martini) MD simulations indicate multiple binding sites for Pf3 in YidC and Nonequilibrium atomistic MD simulations quantified the strength of binding in different binding sites. In particular, the stronger binding with YidC hydrophilic groove and the weaker binding with the cytoplasmic alpha helices were identified using these computational studies. Mutational studies were used to examine some of the findings of the computational studies. The mutated YidC proteins show impaired insertion as expected but they still bind Pf3. The kinetics and thermodynamics of substrate binding using dynamic SMFS.

While this manuscript reports on an extensive study of Pf3-YidC interactions using various complementary techniques, there are major issues with the potential significance of the results.

1. The authors have summarized the free energy landscape of the YidC-mediated binding in Fig. 5e schematically. My main issue with the current manuscript is exactly the fact that there is little new insight provided by this figure. Refs. 9, 10, and 28, which go back to almost a decade ago already provide some similar insights and even estimate duration of the insertion process. It is not clear what is fundamentally different from the mechanisms suggested previously except for some more accurate numbers that the current study suggest. I was not able to identify any major differences between what is already known in the literature and what is proposed here.
2. It has been proposed that the YidC undergoes conformational changes upon substrate binding (e.g., see Ref. 10). This manuscript fails to address the potential importance of these conformational changes and whether or not these conformational changes have any impact on the interpretation of the results observed. Do the authorize observe any evidence of conformational changes in any of their experiments or simulations? Specially since the simulations are often short as compared to the timescales associated with most functionally important events, can we trust the simulations that start from an apo structure and most likely stay close (in terms of YidC conformation) to that structure?

3. The MD simulations attempt to quantify the interactions between YidC and Pf3. The authors state that "the simulations showed Pf3 to bind the two cytoplasmic alpha-helices CH1 and CH2 of YidC in diverse orientations". Do we learn anything from this observation? In other words, do we expect anything other than what is observed? The relatively short MD simulations (compared to 2 ms estimated to be the binding time experimentally) are expected to show various binding events that are not necessarily relevant. Pf3 in real life probably makes contact with YidC several times in different orientations and in different locations but the contact is brief (microseconds) and leaves without truly binding. Such simulations may only provide some starting points (binding poses) but to determine whether they are relevant or not, perhaps free energy calculations need to be performed. The authors fail to perform any free energy calculations to be able to reliably determine not only energetics but also the relevance of the binding poses they have identified computationally.

Despite the major issues discussed above, the manuscript is well-written and well-organized and certainly informative and of interest to the people who are interested in YidC-mediated Sec-independent membrane insertion.

Reviewer #3:

Remarks to the Author:

The work by Laskowski et al investigates the insertion of polypeptides into membranes with the intervention of the YidC system. They use an array of techniques involving single-molecule and fluorescence images, also providing a mechanistic description based on MD simulations. The work pushes the limits of the traditional membrane protein single-molecule experiments incorporating imaging techniques that allow for a more visual interpretation of results. In my opinion the work is very good, but the SMFS part is the weakest part of the work, and the authors should try to incorporate some experiments to make this a relevant work. I am happy to suggest some experiments that are simple and that will elevate the overall quality of the work also supporting their conclusion.

The main problem is that the SMSF traces in figures 1, 2 and Supp6 do not show much, only single events that could be also due to unspecific interactions. Showing reproducibility is an extremely important part of SMFS experiments. Have the authors tried to pull back on the inserted Pf3 peptide and reinsert it? If a single trace shows the exact same event several times, there is no ambiguity. A reproducible single-molecule event would definitely provide a strong evidence for the force measurements. Also, the authors should show representative experimental traces with the mutant forms and compare them with WT results.

The authors suggest that "YidC must strengthen its initial binding to the Pf3 polypeptide to be able to insert the protein into the membrane". They also mention residues in the cytoplasmic hairpin that could provide stabilization upon binding. Have the authors considered mutations of these residues in their experiments? Perhaps, multiple mutations at once could provide a much more clear effect even at short contact times.

Minor comments

Perhaps the authors could use standard error of the mean (SEM) for force measurements statistics.

NCOMMS-21-22422-T "Monitoring the binding and insertion of a transmembrane protein by an insertase" Laskowski et al.

Point-by-Point Response to the Reviewer's Comments

Point-by-Point Response to Reviewer #1

Reviewer #1: Using a variety of sophisticated methods, Lakowski et al. present data that reveal the pathway of pf3 insertion into lipid membranes guided by the YidC insertase. The paper is of fundamental importance and should be published--eventually. The fundamental problems are the organization and presentation of the data. Not until the reader reaches the end of the paper does s/he really come to understand the question being examined, the strategy being used to answer it, and the answer. The interested but non-expert reader is likely to abandon the paper wondering what the point of it is. There are five main figures and 17(!) supplementary figures that are exhausting to read even for experts. In short, the paper is a mess and a nightmare to read. Reading the paper is a slog. One thing that would help is a closing figure that summarizes the essential findings for the non-expert reader.

The paper should be reconsidered after the authors dramatically improve the presentation of the material with the help of a native English speaker.

Authors: Thank you for your positive and constructive comments, which guided us to revise our manuscript. The reviewer comments that our paper is not sufficiently clear in communicating the questions examined, the strategy being used to answer it, and the answer/results. This comment somewhat contrasts the comment of reviewer #2 "*... the manuscript is well-written and well-organized and certainly informative.*" However, we have revised our paper to more clearly communicate the questions examined, the strategy being used to answer the questions, and the answers/results. We have also revised the closing figure to summarize the main findings presented in the manuscript (**Fig. 5e**). We hope that the revised motivation, results, and outcome of our paper are now much better understandable and that the paper is much better to read.

The reviewer also mentions that the "*five main figures and 17(!) supplementary figures are exhausting to read even for experts*". We agree that we present an extensive amount of supplementary figures. However, these figures are necessary to detail the experimental and computational setups (**Supplementary Fig. S1, S2, S5, S6**), to detail the experimental data, which analyses are shown in the main figures (**Supplementary Fig. S7 – S10, S12**), and to show the essential controls (**Supplementary Fig. S3 – S5**). We thus think that the substantial amount of supplementary data are helpful to better understand the experimental and theoretical setups, the data and the controls supporting the main findings of our manuscript. Nevertheless, in our revision we have reduced the Supplementary Figures to 15.

In summary, we have revised our paper to improve its readability (motivation, strategy, answers), included a considerably revised closing figure, and reduced the Supplementary

Figures. For details please see our considerably revised Manuscript (Abstract, Introduction, Results, Discussion and Supplementary).

Point-by-Point Response to Reviewer #2

Reviewer #2: The manuscript by Laskowski et al. provides mechanistic details on the binding and insertion of a single-pass transmembrane protein (Pf3) mediated by YidC, a bacterial insertase. The authors use a combination of various biophysical, computational, and biochemical techniques to decipher the Pf3 binding and insertion process by YidC. Using a combination of AFM and SMFS, the lifetime and strength associated with binding events were characterized. The single-molecule assays show that the binding strength between YidC and Pf3 depends on the contact time. They conclude that Pf3 binds YidC within less than 2 ms and inserts into the membrane within 52 ms. Their coarse-grained (Martini) MD simulations indicate multiple binding sites for Pf3 in YidC and Nonequilibrium atomistic MD simulations quantified the strength of binding in different binding sites. In particular, the stronger binding with YidC hydrophilic groove and the weaker binding with the cytoplasmic alpha helices were identified using these computational studies. Mutational studies were used to examine some of the findings of the computational studies. The mutated YidC proteins show impaired insertion as expected but they still bind Pf3. The kinetics and thermodynamics of substrate binding using dynamic SMFS.

While this manuscript reports on an extensive study of Pf3-YidC interactions using various complementary techniques, there are major issues with the potential significance of the results.

Authors: We thank the reviewer for the positive and constructive comments, which guided us to revise our manuscript. Below, we address point by point the comments from the reviewer.

Reviewer #2: 1) The authors have summarized the free energy landscape of the YidC-mediated binding in Fig. 5e schematically. My main issue with the current manuscript is exactly the fact that there is little new insight provided by this figure. Refs. 9, 10, and 28, which go back to almost a decade ago already provide some similar insights and even estimate duration of the insertion process. It is not clear what is fundamentally different from the mechanisms suggested previously except for some more accurate numbers that the current study suggest. I was not able to identify any major differences between what is already known in the literature and what is proposed here.

Authors: The reviewer comments that from Fig. 5e “it is not clear what is fundamentally different from the mechanisms suggested previously except for some more accurate numbers that the current study suggest”. Thank you we have revised Fig. 5e for a better understanding. The revised Fig. 5e describes the schematic free-energy landscape of the YidC-mediated Pf3 insertion process at improved detail, and shows distinct states with the first state being the substrate binding by the cytoplasmic α -helical hairpins. This initial binding, which is mainly facilitated by the cytoplasmic α -helices of YidC, has a long lifetime of ≈ 33 s such as needed to prevent the YidC-Pf3 complex to dissociate. The fact that the YidC-Pf3 complex describing the initial binding of the substrate can show many different conformations of long lifetime suggests the cytoplasmic α -helices of YidC to work like an efficient ‘flytrap’ to catch (bind) the substrate from the cytosol. In a next step, the Pf3 polypeptide with the help of the cytoplasmic

α -helices is guided to the hydrophilic groove. After binding the hydrophilic groove, the Pf3 reaches the free-energy valley of the membrane-inserted state within 52 ms. This insertion is supported by electrostatic attractions between negatively-charged residues of the Pf3 N-terminus and R366 of the hydrophilic groove of YidC. However, the cytoplasmic α -helical hairpin also contributes to this process. Once the Pf3 polypeptide is inserted and folded into the membrane, the N-terminal region will move from the groove-bound state to the periplasm¹⁵. Compared to the free-energy valley describing the initial substrate-binding, this second valley of the membrane inserted state is much narrower thus providing Pf3 less conformational variability, such as described for transmembrane α -helices^{27,34}. After this insertion and folding the Pf3 can dissociate from YidC.

Indeed, previously published papers suggested different insertion steps including temporary interactions formed between Pf3 and the hydrophilic groove of YidC^{1,2} or the transmembrane α -helices TMH3 and TMH5 of YidC^{3,4}. A role of the cytoplasmic α -helical hairpins has not been assigned in these publications. To our best knowledge, only the publication from Winterfeld *et al.*⁵ provided information about the time of the YidC-mediated membrane protein insertion process. Despite the difference in the biophysical techniques used, we measured comparable times needed for YidC to insert Pf3 into lipid membranes as reported by Winterfeld *et al.* However, the major novel result depicted in **Fig. 5e** is that our measurements detail two steps of the YidC-mediated membrane protein insertion process. The first step describing the initial substrate binding is established within ≈ 2 ms by the cytoplasmic α -helical hairpins. The second, much longer step describing the insertion of the substrate into the membrane is completed within ≈ 52 ms. **Fig. 5e** also summarizes the kinetics (lifetimes) of the substrate binding state and of the substrate insertion state and thus quantifies additional parameters of the YidC-mediated membrane protein binding and insertion process. In addition, the free-energy landscape parameters given in **Fig. 5e** quantitate the conformational variabilities of the binding and of the insertion steps. Particularly, the high conformational variability of the initially bound YidC-Pf3 complex contrasts the more rigid, membrane inserted state of Pf3, both of which have not been reported before.

In summary, we hope that the considerably revised **Fig. 5e** and the revised discussion describe the novelty of our mechanistic findings now more clearly (see revised Manuscript, section Introduction, section Results, section Discussion, and revised **Fig. 5e**).

Reviewer #2: 2) It has been proposed that the YidC undergoes conformational changes upon substrate binding (e.g., see Ref. 10). This manuscript fails to address the potential importance of these conformational changes and whether or not these conformational changes have any impact on the interpretation of the results observed. Do the authorize observe any evidence of conformational changes in any of their experiments or simulations? Specially since the simulations are often short as compared to the timescales associated with most functionally important events, can we trust the simulations that start from an apo structure and most likely stay close (in terms of YidC conformation) to that structure?

Authors: The reviewer comments that our manuscript fails to address the potential importance of the conformational changes YidC undergoes upon substrate binding, which have been proposed in several publications.^{6,7} The reviewer further asks whether we observe

any evidence of conformational changes in our experiments or simulations, especially since the simulations are often short as compared to the timescales associated with most functionally important events. Thank you for your constructive comment. We would kindly like to mention, that protein dynamics, e.g. conformational changes upon activation of G-protein coupled receptors can take place at microsecond timescales.⁸ Our MD simulations, which also approach microsecond time scales (**Supplementary Table 2**), observe conformational changes of YidC upon substrate binding and insertion, which we detail in the following. The numerous insight provided by our MD simulations, which could contribute to a paper on their own, are summarized below for the main conformational changes between (i) an apo simulation of YidC, (ii) YidC-Pf3 complexes in which Pf3 is bound to the cytoplasmic α -helices CH1 and CH2, and (iii) YidC-Pf3 complexes in which Pf3 is bound to the hydrophilic groove of YidC. The largest observed conformational changes of YidC correspond to the movement of the cytoplasmic α -helices CH1 and CH2 (**Fig. R1**), the opening of the hydrophilic groove (**Fig. R2**), and the mobility of the amphipathic helix EH1 and periplasmic loop PL2 (**Fig. R3**). Moreover, we observe that Pf3 binding alters the conformation of the short periplasmic loop PL3 of YidC. Thus, taken together, our simulations indicate conformational changes of different structural regions of YidC upon Pf3 binding to both the cytoplasmic α -helices as well as to the hydrophilic groove. Furthermore, our simulation results are supported by our experimental results that approximate the conformational variability of the YidC-Pf3 complexes. The experiments observe that YidC-Pf3 complexes representing the initially bound state of Pf3 show a large conformational variability (e.g. large x_u value), which correlates to the diverse structures of CH1/CH2 bound complexes observed in the spontaneous binding simulations (**Supplementary Fig. S6, S7**). In contrast, the experimentally observed membrane-inserted state of Pf3 shows a much smaller conformational variability (e.g. small x_u value), which is in agreement with the similar conformations YidC forms with Pf3 *via* conserved interactions in the hydrophilic groove (**Supplementary Fig. S8**). We have now revised our paper to more clearly report the results from simulations and experiments and how they relate to the importance of conformational changes of YidC (see revised Manuscript, sections Introduction and Discussion). However, we did not include the additional analysis (**Fig. R1, R2, and R3**) into our revision since reviewer #1 found that the number of our Supplementary Figures is too extensive. Thus, if reviewer #2 and the editor explicitly wish so we would integrate **Fig. R1, R2, and R3** into the Supplementary Information of our manuscript.

Figure R1. Positioning of cytoplasmic α -helices CH1 and CH2 upon Pf3 binding as obtained from MD simulations. (a) The conformation of the cytoplasmic α -helices CH1 and CH2 was monitored by measuring the tilt angle between the backbone of CH1 (black, tilt angle in magenta) and CH2 (grey, tilt angle in green) and the membrane plane (black line). (b) Bar charts of tilt angles estimated for the apo YidC (no substrate, black), for Pf3 bound to the cytoplasmic α -helices CH1 and CH2 of YidC (grey), and for Pf3 bound to the hydrophilic groove of YidC (blue). The results show that the tilt angle of CH1 and CH2 increases in hydrophilic groove-bound YidC-Pf3 complexes by $\approx 5^\circ$ compared to apo YidC and varies only little. In YidC-Pf3 complexes where Pf3 is bound to CH1/CH2, the tilt angles vary more, as compared to the hydrophilic groove-bound complexes. Interestingly, the tilt of the individual cytoplasmic α -helices is decoupled if Pf3 is bound to CH1/CH2. While the CH1 accommodates the same tilt angle as in the apo structure, the CH2 tilt increases on average by almost 10° . Values represent averages over different simulations after exclusion of initial 100 ns for equilibration. Error bars denote SEM over different simulations (one apo YidC simulation, six simulations of Pf3 bound to CH1/CH2 of YidC, and eight simulations of Pf3 bound to the hydrophilic groove of YidC).

Figure R2. Opening of the hydrophilic groove of YidC upon Pf3 binding as obtained from MD simulations. (a) The opening of the hydrophilic groove of YidC was monitored by measuring distance between C_α atoms of TM3 and TM5 in three different positions. The three positions are residues C423-M495 close to cytoplasmic α -helices CH1 and CH2 (black, entry in (b)), residues L426-M498 in the middle of the hydrophilic groove (magenta, mid in (b)) and between residues F433-F505 located deeply in the membrane at the end of the hydrophilic groove (yellow, exit in (b)). For simplicity Pf3 is not shown in the image. (b) The distances between TM3 and TM5 characterize the opening for the apo YidC (no substrate, black), for Pf3 bound to the cytoplasmic α -helices CH1 and CH2 of YidC (grey), and for Pf3 bound to the hydrophilic groove of YidC (blue). The results show that when Pf3 binds to CH1/CH2, the distance between the entry residues of YidC increases by ≈ 0.3 nm, whereas the distances between mid and exit residues pairs is less affected. In contrast, Pf3 bound to the hydrophilic groove considerably increases the distance of all three residue pairs entry, mid, and exit of TM3 and TM5 by 0.6 nm, 0.5 nm, and 0.35 nm, respectively. The insertion of Pf3 between TM3 and TM5 opens the hydrophilic groove similarly to recently observed opening of the transmembrane channel of the SecYEG translocase.⁹ The analysis includes one apo YidC simulation, six simulations of Pf3 bound to CH1/CH2 of YidC, and eight simulations of Pf3 bound to the hydrophilic groove of YidC).

Figure R3. Root mean square deviations (RMSD) and root mean square fluctuations (RMSF) of C α atoms averaged over periplasmic regions of YidC. (a) Highlighting the structural regions of YidC analyzed for RMSD: periplasmic loops PL1 (residues V351-W354, magenta), PL2 (residues S443-Y465, yellow), and PL3 (residues W508-P510, black), and the amphipathic helix EH1 (residues V328-F350, blue). For simplicity Pf3 is not shown. (b) The analysis of the MD simulations (apo YidC (no substrate, black), Pf3 bound to the cytoplasmic α -helices CH1 and CH2 of YidC (grey), and Pf3 bound to the hydrophilic groove of YidC (blue)) shows that Pf3 binding influences the structure of EH1, PL2 and PL3 (labeled on a structure on the left), as indicated by larger RMSD from the crystal structure 6AL2 as compared to the apo simulation of YidC but not the structure of PL1. At the same time, Pf3 binding increases the mobility of EH1 and PL2 of YidC as indicated by increased RMSF values. The presented results are in agreement with the experimental quantification of the mobility of the periplasmic regions of YidC.⁶ The analysis includes one YidC apo simulation, six simulations of Pf3 bound to CH1/CH2 of YidC, and eight simulations of Pf3 bound to the hydrophilic groove of YidC).

Reviewer #2: 3) The MD simulations attempt to quantify the interactions between YidC and Pf3. The authors state that "the simulations showed Pf3 to bind the two cytoplasmic α -helices CH1 and CH2 of YidC in diverse orientations". Do we learn anything from this observation? In other words, do we expect anything other than what is observed? The relatively short MD simulations (compared to 2 ms estimated to be the binding time experimentally) are expected to show various binding events that are not necessarily relevant. Pf3 in real life probably makes contact with YidC several times in different orientations and in different locations but the contact is brief (microseconds) and leaves without truly binding. Such simulations may only provide some starting points (binding poses) but to determine whether they are relevant or not, perhaps free energy calculations need to be performed. The authors fail to perform any free energy calculations to be able to reliably determine not only energetics but also the relevance of the binding poses they have identified computationally.

Authors: The referee argues that the diverse binding conformations observed in our simulations, in which Pf3 localizes to the cytoplasmic α -helices CH1 and CH2, may or may not be relevant for the insertion. We kindly agree that some of the spontaneous binding events observed from coarse-grained (CG) simulations at time ranges of tens of μ s are likely initial binding events of Pf3 to YidC, which may not be relevant for the full insertion process of the substrate. Thus, in our manuscript we do not claim whether and how these conformations may be of relevance for the insertion process. It may, however, be assumed that the diverse initial binding conformations over the time range of several ms (we measure an insertion time of ≈ 52 ms) migrate into conformations that allow YidC to guide the Pf3 polypeptide towards the hydrophilic groove. Therefore, we take samples of those initial binding conformations, re-equilibrate them atomistically and perform *in silico* pulling experiments at atomistic resolution in order to compare them to Pf3 adsorbed to the bilayer and to Pf3 bound in the hydrophilic groove of YidC (**Fig. 3, Supplementary Fig. S7, S8, S10**). Interestingly, we observe that some of the initial binding conformations establish weak interactions comparable to the interaction strengths of Pf3 adsorbed to the bilayer, whereas other conformations establish interactions being as strong as Pf3 bound in the hydrophilic groove (**Fig. 3d**). However, as we do not want to overinterpret these results, we do not suggest which among the initial binding conformations are more relevant than others. We agree with the referee that performing free-energy calculations could help to identify the relevance of those binding conformations. However, we mechanically pull Pf3 by more than 20 nm to separate it from YidC and Pf3 consists of 44 amino acids contributing to a large number of degrees of freedom. Currently, it is thus computationally not feasible to atomistically simulate such high variabilities of conformations over the full pulling distance and to reliably calculate the potential of mean force (PMF). We kindly remind that to estimate the PMF from one pulling path of only one binding conformation would require > 200 atomistic MD simulations, each being simulated for multiple μ s. Also, performing much shorter ($\ll \mu$ s time spans) MD simulations along individual pulling pathways would considerably distort the free-energy estimations of the different binding conformations, because the much higher pulling speed would force the simulated system far from equilibrium. Interpreting such distorted systems (far from equilibrium) would lead to draw wrong conclusions, which we would like to avoid.

Reviewer #2: 4) Despite the major issues discussed above, the manuscript is well-written and well-organized and certainly informative and of interest to the people who are interested in YidC-mediated Sec-independent membrane insertion.

Authors: Thank you, we appreciate the positive and constructive feedback from the reviewer.

Point-by-Point Response to Reviewer #3

Reviewer #3: The work by Laskowski et al investigates the insertion of polypeptides into membranes with the intervention of the YidC system. They use an array of techniques involving single-molecule and fluorescence images, also providing a mechanistic description based on MD simulations. The work pushes the limits of the traditional membrane protein single-molecule experiments incorporating imaging techniques that allow for a more visual interpretation of results. In my opinion the work is very good, but the SMFS part is the weakest part of the work, and the authors should try to incorporate some experiments to make this a relevant work. I am happy to suggest some experiments that are simple and that will elevate the overall quality of the work also supporting their conclusion.

Authors: Thank you for your positive and constructive comments, which have guided us to revise our manuscript. Below we have answered every specific comment of the reviewer in detail.

Reviewer #3: The main problem is that the SMSF traces in figures 1, 2 and Supp6 do not show much, only single events that could be also due to unspecific interactions. Showing reproducibility is an extremely important part of SMFS experiments. Have the authors tried to pull back on the inserted Pf3 peptide and reinsert it? If a single trace shows the exact same event several times, there is no ambiguity. A reproducible single-molecule event would definitely provide a strong evidence for the force measurements. Also, the authors should show representative experimental traces with the mutant forms and compare them with WT results.

Authors: The reviewer comments that the single-molecule force spectroscopy (SMFS) traces in **Fig. 1, 2** and **Supplementary Fig. S5** do not show much, only single events that could be also due to unspecific interactions. He/she further points out that it is of importance to show the reproducibility of the SMFS experiments. We fully agree with the reviewer. To be able to separate specific from unspecific interactions is prerequisite to analyze SMFS data correctly. It is also correct that the presented force curves show only single unbinding / adhesion event. However, such single force peaks are expected when probing single interactions¹⁰ or upon mechanically extracting single transmembrane α -helices from the membrane¹¹. To be able to separate specific from unspecific interactions, the SMFS experiments should be designed in a way that only single unbinding events are detected and can be clearly discerned from possible unspecific interactions.¹² Accordingly, we have designed our SMFS experiments to be able to discern specific from unspecific interactions detected in the force curves. In our design, which is shown in **Supplementary Fig. S2**, the Pf3 polypeptide is chemically tethered to the AFM tip *via* a PEG₂₇-linker. In the stretched conformation this PEG₂₇-linker is ≈ 9 nm long. At the end of the linker we have covalently attached the C-terminal end of the Pf3 polypeptide. If this Pf3 polypeptide interacts with YidC, then an adhesive interaction of the Pf3 polypeptide and YidC is detected at a certain pulling distance from the contact point of AFM tip and membrane (or YidC in residing in the membrane). In our experimental setup this distance corresponds to the stretched length of PEG₂₇-linker and the interaction site of Pf3 polypeptide. In contrast, unspecific interactions between AFM tip and sample occur at the contact point of tip and sample, which is at pulling distances $\approx 0 - 2$ nm.^{10,13,14} Thus, applying a distance filter of

≈ 5 nm separates the unspecific ($\ll 5$ nm) from specific (≥ 5 nm) adhesive force peaks. The force curves displayed in **Fig. 2** and **Supplementary Fig. S5** show adhesive peaks at distances corresponding to the linker length (length of PEG₂₇-linker and of Pf3 polypeptide). Therefore, one can conclude that the force peaks detected at distances ≥ 5 nm are due to specific interactions of the P3 polypeptide with YidC. In **Fig. 1**, the AFM tip is kept at a distance of ≈ 5 –10 nm from the YidC membrane. At this distance no unspecific interactions occur. Thus, if the Pf3 polypeptide, which has been tethered to the AFM tip *via* a PEG₂₇ linker, interacts with the membrane these interactions are detected as specific binding events. We hope that we could better explain the criteria we applied to detect and select specific interactions in our force spectroscopy experiments. These experimental (linker system) and data analysis criteria are standard to the field^{10,13,14}. To avoid confusion of the reader we have revised our Manuscript to explain these criteria at better detail (see revised Results and Methods sections ‘Atomic force microscopy (AFM) tip functionalization’ and ‘Analysis of force-distance and force-time curves’).

The reviewer suggests an experiment to extract Pf3 and to reinserted it into the membrane in order to prove a repetitive unfolding (and folding) behavior. For example, in our experiments displayed in **Fig. 2**, we have used the same Pf3 functionalized AFM tip to repetitively probe the unfolding (and extraction) and the insertion of Pf3 by YidC. Below we show force curves detected with the same Pf3 functionalizing the AFM tip and showing the repetitive insertion, unfolding and extraction, and insertion of Pf3 from YidC (**Fig. R4**). In addition, we also provide controls in which we examined the unspecific adhesion of Pf3 to phospholipid membranes and to purple membrane (**Supplementary Fig. S3**). In these control experiments the lipid membranes or purple membranes were co-adsorbed with YidC membranes, so that we could directly probe both the unspecific (lipid or purple membranes) and specific (YidC membranes) interactions of Pf3 in the same experiment (with the same functionalized AFM tip). The experiments show the specific binding and insertion of Pf3 to YidC membranes but not to lipid membranes or to purple membranes.

Figure R4. Repetitive binding of the Pf3 coat protein to a membrane with reconstituted YidC. The FD curves have been recorded with an AFM tip that has been functionalized with Pf3 as described (**Fig. 1**, **Supplementary Fig. S2**) to detect specific binding events to YidC. As described in **Fig. 2**, single binding events were detected in the SMFS mode, which had been simultaneously conducted while recording the FD-AFM topograph. The representative FD curves of the left and of the right columns, each of

which having been detected using the same Pf3-functionalized AFM tip, show repetitive single unbinding (adhesion) events detected at tip-membrane distances corresponding to the length of the PEG₂₇-linker and Pf3 polypeptide tethered to the AFM tip. FD curves were taken from the data set presented in **Fig. 2c** collected at 52 ms contact time. At contact times of 52 ms YidC has completed the insertion of Pf3 into the membrane. Please note that each functionalized AFM tip may contain multiple covalently bound Pf3 molecules, which, depending on their vertical attachment to the tip, can show adhesive interactions, such as seen for the last FD curve (bottom) of the right column.

The reviewer also suggests to show representative experimental force curves with the mutant forms and compare them with WT results. In our revision we now show representative experimental force curves with the mutant YidC forms so that they can be compared with the representative experimental force curves obtained from wt YidC (**Fig. R5**). However, we kindly note that individual force curves characterizing the unbinding of a single Pf3 polypeptide from wt YidC or mutant YidC are essentially indistinguishable, as they both show single unbinding force peaks. Based on the nature of the single-molecule experiments the magnitude of these unbinding force peaks varies stochastically. Thus, only after analyzing the average and distribution of many hundreds of (un-)binding forces recorded for multiple experimental condition, we can identify differences between Pf3 binding to wt YidC and Pf3 binding to mutant R366E YidC or to mutant Δ CH2 YidC.

Figure R5. Force curves with Pf3 unbinding from YidC. Force curves show Pf3 unbinding from **(a)** wt YidC (purple), **(b)** Δ CH2 YidC (grey) and **(c)** R366E YidC (orange). Force curves reveal similar features binding to wt YidC and two YidC mutants. Force curves were taken from data presented in **Fig. 2c, 4c,d** collected at 2 ms contact time.

In summary, we hope that we could fully address the comments in our detailed response. However, we did not include the additional force curves (**Fig. R4** and **R5**) into our revision since reviewer #1 found that the number of our Supplementary Figures is too extensive. Thus, if reviewer #3 and the editor explicitly wish so we would integrate **Fig. R4** and **R5** into the Supplementary Information of our manuscript.

Reviewer #3: The authors suggest that "YidC must strengthen its initial binding to the Pf3 polypeptide to be able to insert the protein into the membrane". They also mention residues in the cytoplasmic hairpin that could provide stabilization upon binding. Have the authors considered mutations of these residues in their experiments? Perhaps, multiple mutations at once could provide a much more clear effect even at short contact times.

Authors: Thank you for your comment. Indeed, based on the molecular dynamics (MD) simulations we have identified potentially critical residues within cytoplasmic α -helices CH1 and CH2. Based on this observation, we have examined the effect of mutating the positively charged residues R384E and K401E of YidC (**Fig. R6, now included as new Supplementary Fig. S15**). Both YidC mutants were studied by fluorescence correlation spectroscopy (FCS) and Förster resonance energy transfer (FRET), as described for wt and mutant YidC in the manuscript. Because R384E YidC showed an insertion efficiency similar to wt YidC (**Fig. R6a, R6b**), we decided not to study the Pf3 binding of this mutant by SMFS. On the other hand, K401E YidC showed a drastically impaired Pf3 insertion (**Fig. R6c**) and shifted the Pf3 binding site (**Fig. R6d**). Unfortunately, we could not manage to collect enough (un-)binding events by SMFS to be able to reconstruct reliable (un-)binding force distributions, which was likely because of the decreased probability of K401E YidC to bind the Pf3 substrate. A more detailed characterization of these and other residues, which we briefly discuss in our revised Manuscript (see Discussion and **new Supplementary Fig. S15**), could be a focus of future studies.

Figure R6, now included as Supplementary Fig. S15. Single mutations in the cytoplasmic helices CH1 and CH2 of YidC affecting the binding and insertion of Pf3. (a) Pf3 insertion into R384 (yellow) and wt (purple) YidC proteoliposomes as measured by FCS. The Atto520 dye attached to the N-terminal end of Pf3-3C (**Methods**) is quenched outside proteoliposomes and bursts fluorescence upon translocation *via* YidC into proteoliposomes. **(b)** FRET histograms of Pf3 binding to wild-type (wt) YidC S405C and R384E YidC S405C. FRET efficiencies calculated for each individual burst are shown for wt YidC S405C (purple) and R384E YidC S405C (yellow). Förster distances were calculated based on the calculated average FRET efficiency (E-FRET) values for single bursts of fluorescence. **(c)** Pf3 insertion into wt YidC S405C (purple) and K401E YidC S405C (green) proteoliposomes as measured by FCS. Data points represent means from 35 measurements and error bars sd. **(d)** FRET histograms of Pf3 binding to wt YidC S405C and K401E YidC S405C.

Reviewer #3: Minor comments

Perhaps the authors could use standard error of the mean (SEM) for force measurements statistics.

Authors: We agree that the SEM would be helpful to estimate the accuracy of the mean values. However, by using standard deviation (SD) we have chosen to highlight the high variability of the measurements, which likely indicates that YidC and Pf3 establish multiple bonds / interactions at each contact time. In contrast a high number of measurements would cause SEM to be small thereby hiding this variability.

References

- 1 Kumazaki, K. *et al.* Structural basis of Sec-independent membrane protein insertion by YidC. *Nature* **509**, 516-520, doi:10.1038/nature13167 (2014).
- 2 Chen, Y., Soman, R., Shanmugam, S. K., Kuhn, A. & Dalbey, R. E. The role of the strictly conserved positively charged residue differs among the Gram-positive, Gram-negative and chloroplast YidC homologs. *J. Biol. Chem.* **289**, 35656-35667, doi:10.1074/jbc.M114.595082 (2014).
- 3 He, H., Kuhn, A. & Dalbey, R. E. Tracking the Stepwise Movement of a Membrane-inserting Protein In Vivo. *J. Mol. Biol.* **432**, 484-496, doi:10.1016/j.jmb.2019.10.010 (2020).
- 4 Yu, Z., Koningstein, G., Pop, A. & Luirink, J. The Conserved Third Transmembrane Segment of YidC Contacts Nascent Escherichia coli Inner Membrane Proteins. *J. Biol. Chem.* **283**, 34635-34642, doi:10.1074/jbc.M804344200 (2008).
- 5 Winterfeld, S., Ernst, S., Borsch, M., Gerken, U. & Kuhn, A. Real time observation of single membrane protein insertion events by the Escherichia coli insertase YidC. *PLoS One* **8**, e59023, doi:10.1371/journal.pone.0059023 (2013).
- 6 Imhof, N., Kuhn, A. & Gerken, U. Substrate-Dependent Conformational Dynamics of the Escherichia coli Membrane Insertase YidC. *Biochemistry* **50**, 3229-3239, doi:10.1021/bi1020293 (2011).
- 7 Kedrov, A. *et al.* Structural Dynamics of the YidC:Ribosome Complex during Membrane Protein Biogenesis. *Cell Rep.* **17**, 2943-2954, doi:<https://doi.org/10.1016/j.celrep.2016.11.059> (2016).
- 8 Dror, R. O. *et al.* Activation mechanism of the β 2-adrenergic receptor. *Proc. Natl. Acad. Sci.* **108**, 18684, doi:10.1073/pnas.1110499108 (2011).
- 9 Mercier, E., Wang, X., Maiti, M., Wintermeyer, W. & Rodnina, M. V. Lateral gate dynamics of the bacterial translocon during cotranslational membrane protein insertion. *Proceedings of the National Academy of Sciences* **118**, e2100474118, doi:10.1073/pnas.2100474118 (2021).
- 10 Dufrene, Y. F., Martinez-Martin, D., Medalsy, I., Alsteens, D. & Muller, D. J. Multiparametric imaging of biological systems by force-distance curve-based AFM. *Nat. Methods* **10**, 847-854, doi:10.1038/nmeth.2602 (2013).
- 11 Engel, A. & Gaub, H. E. Structure and Mechanics of Membrane Proteins. *Annu. Rev. Biochem.* **77**, 127-148, doi:10.1146/annurev.biochem.77.062706.154450 (2008).
- 12 Johnson, K. C. & Thomas, W. E. How Do We Know when Single-Molecule Force Spectroscopy Really Tests Single Bonds? *Biophys. J.* **114**, 2032-2039, doi:<https://doi.org/10.1016/j.bpj.2018.04.002> (2018).
- 13 Hinterdorfer, P. & Dufrêne, Y. F. Detection and localization of single molecular recognition events using atomic force microscopy. *Nat. Methods* **3**, 347-355, doi:10.1038/nmeth871 (2006).
- 14 Müller, D. J. & Dufrêne, Y. F. Atomic force microscopy as a multifunctional molecular toolbox in nanobiotechnology. *Nat. Nanotechnol.* **3**, 261-269, doi:10.1038/nnano.2008.100 (2008).

Reviewers' Comments:

Reviewer #2:

Remarks to the Author:

The authors have addressed my concerns.

Reviewer #3:

Remarks to the Author:

The authors have done a relevant effort to address my concerns. I still believe that traces showing a single peak are dangerous, although the ones reported look quite similar in extension and force. I would encourage the authors to include this new data in the Supp. I recommend publication

NCOMMS-21-22422A "Monitoring the binding and insertion of a transmembrane protein by an insertase" Laskowski et al.

Point-by-Point Response to the Reviewer's Comments

Point-by-Point Response to Reviewer #2

Reviewer #2: The authors have addressed my concerns.

Authors: We thank the reviewer for the positive and constructive comments, which guided us to revise our manuscript.

Point-by-Point Response to Reviewer #3

Reviewer #3: The authors have done a relevant effort to address my concerns. I still believe that traces showing a single peak are dangerous, although the ones reported look quite similar in extension and force. I would encourage the authors to include this new data in the Supp. I recommend publication

Authors: Thank you for your positive and constructive comments, which have guided us to revise our manuscript. As recommended by the reviewer we have included the data showing single force curves in the supplementary information (**new Supplementary Fig. 16**).